# Probing Equivariance and Symmetry Breaking in Convolutional Networks

**Sharvaree Vadgama**[1,2,*]     **Mohammad Mohaiminul Islam**†[3]     **Domas Buracas**†[2†]
**Christian Shewmake**[2,4]     **Artem Moskalev**[5]     **Erik J. Bekkers** [1]
[1]AMLab, University of Amsterdam    [2] New Theory AI
[3] QurAI, University of Amsterdam    [4] UC Berkeley   [5] Independent Researcher

## Abstract

In this work, we explore the trade-offs of explicit structural priors, particularly group-equivariance. We address this through theoretical analysis and a comprehensive empirical study focusing on point clouds. To enable controlled and fair comparisons, we introduce `Rapidash`, a unified group convolutional architecture that allows for different variants of equivariant and non-equivariant models. Our results suggest that more constrained equivariant models outperform less constrained alternatives when aligned with the geometry of the task, and increasing representation capacity does not fully eliminate performance gaps. We see improved performance of models with equivariance and symmetry-breaking through tasks like segmentation, regression, and generation across diverse datasets. Explicit *symmetry breaking* via geometric reference frames consistently improves performance, while *breaking equivariance* through geometric input features can be helpful when aligned with task geometry. Our results provide task-specific performance trends that offer a more nuanced way for model selection. Code available at `github.com/Sharvaree/EquivarianceStudy`.

## 1   Introduction

There is an ongoing debate about the necessity of explicit structural priors, particularly group-equivariance. A growing line of work argues that strict equivariance may over-constrain a model and limit its scalability, and that increasing model capacity and training data [Qu and Krishnapriyan, 2024] can compensate for the lack of built-in symmetry. This perspective is supported by models like AlphaFold [Abramson et al., 2024], which train equivariance by data augmentation. In contrast, several recent works highlight the limitations of learning equivariance from data alone. Moskalev et al. [2023] show that learned symmetries can be unreliable and rapidly degrade out-of-distribution. Gruver et al. [2024] demonstrates that aliasing and nonlinearities are primary contributors to layerwise equivariance loss, and that data and training scale can have a greater influence than architectural design in learning equivariances.Petrache and Trivedi [2023] demonstrate that even partial symmetry alignment can improve generalization. Theoretical results by Perin and Deny [2024] further highlight that learning equivariance is fundamentally limited when class orbits are sparse or poorly separated. Given these competing perspectives, it remains unclear under what conditions equivariance leads to tangible benefits, and when unconstrained models may be preferable, highlighting the need for a deeper understanding of this trade-off.

To address this debate, we systematically investigate the impact of equivariant and non-equivariant models across a range of learning tasks involving geometry. We begin by formalizing five core research questions: *(i)* how kernel constraints affect expressivity and generalization, *(ii)* whether

---

*Corresponding author: <`sharvaree.vadgama@gmail.com`>
†equal contribution

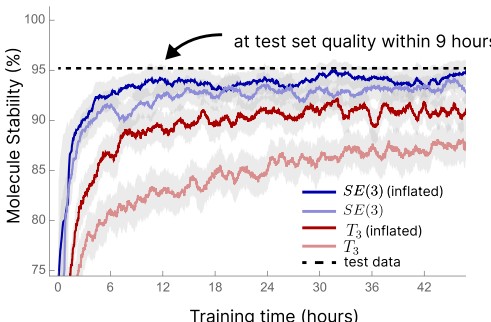

Figure 1: We plot molecular stability and training time for $SE(3)$ and $T_3$ equivariant models with a hidden dimension of 256 / 512(inflated) for the molecular generation task on QM9. Inflated models have representations with higher hidden dimension, hence increased capacity.

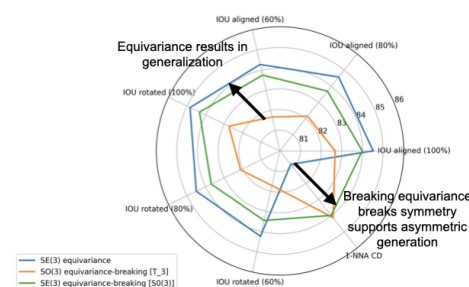

Figure 2: Comparison of `Rapidash` model variants with different equivariance-breaking on Shapenet part segmentation and generation. (X%) indicates what portion of the training dataset the model was trained on. We also evaluate on aligned and rotated versions of the test set.

increased representation capacity can close the performance gap between more and less constrained models, *(iii)* how dataset size influences performance, *(iv)* when pose-informed symmetry breaking improves equivariant models, and *(v)* how different forms of equivariance breaking affect performance.

We approach these questions through both theoretical and empirical analysis. Theoretically, we show that equivariant models decompose into symmetry-preserving and symmetry-breaking components, and provide formal results that link generalization to pose entropy and inductive bias alignment. To conduct systematic empirical analysis, we introduce `Rapidash`, a unified group convolutional architecture that enables fine-grained control over equivariance constraints, input/output representations, and symmetry- and equivariance-breaking mechanisms. This design supports systematic comparisons across a wide range of model variants with varying degrees of equivariance. Experiments on molecular property prediction and generation, 3D part segmentation and shape generation, and motion prediction show that more constrained equivariant models outperform less constrained alternatives when aligned with task geometry. Increasing representation capacity helps both strongly and weakly constrained models, but does not fully eliminate performance gaps (Fig. 1). Models with stronger equivariance are more data-efficient. Explicit symmetry breaking via geometric reference frames consistently improves performance, while breaking equivariance by introducing geometric input features (e.g., coordinates as scalars) can be helpful when aligned with task geometry (Fig. 2). These results provide concrete answers to the research questions and clarify the conditions under which different degrees of equivariance are most effective. `Rapidash` with equivariance-breaking and symmetry-breaking achieves state-of-the-art performance on QM9 generation, property prediction, and CMU motion prediction tasks.

The contributions of this paper can be summarized as:

- **Theoretical analysis**: We provide formal results showing when equivariance and symmetry breaking offer provable advantages in expressivity and generalization.

- **Empirical study**: We systematically study the impact of various equivariant and unconstrained models across multiple learning tasks on point clouds, covering molecular property prediction and generation, 3D part segmentation and shape generation, and motion prediction.

- **Unified architecture**: We introduce `Rapidash`, a scalable and modular group convolutional architecture. This design supports multiple symmetry groups, input/output variants, allowing for equivariance-breaking and symmetry-breaking options.

- `Rapidash` achieves **state-of-the-art** performance on QM9 *molecule generation* and CMU *motion prediction*, relative to recent methods from literature.

The remainder of the paper is organized as follows. Section 2 reviews relevant background. Sec. 3 introduces theory, the research questions and `Rapidash`. Sec. 4 discusses related work. Experimental

results are presented in Sec. 5. Finally, Sec. 6 addresses each research question, summarizes key takeaways on when to prefer equivariant or unconstrained models.

## 2 Background

Our study of equivariant deep learning on 3D point clouds investigates the impact of varying linear layer within convolutional architectures (which alternate linear layers with non-linear operations, e.g., activations), while holding other architectural components fixed to isolate the influence of equivariance. We consider a 3D point cloud $\mathcal{X} = \{\mathbf{x}_1, \ldots, \mathbf{x}_N\} \subset \mathbb{R}^3$ of $N$ points with associated feature fields $f : \mathcal{X} \to \mathbb{R}^C$. Assuming neighborhood sets $\mathcal{N}(i)$ define connectivity for each point $\mathbf{x}_i$ (indexed by $\mathcal{V} = \{1, \ldots, N\}$), the general form of a linear layer for such feature fields is given by

$$[Lf](\mathbf{x}_i) = \sum_{j \in \mathcal{N}(i)} k(\mathbf{x}_i, \mathbf{x}_j) f(\mathbf{x}_j), \tag{1}$$

where the kernel $k(\mathbf{x}_i, \mathbf{x}_j) \in \mathbb{R}^{C' \times C}$ maps point features to $C'$ output channels.

In works such as [Cohen et al., 2019, Bekkers, 2019], constraining linear layers (1) to be equivariant is shown to yield group convolutions. For instance, translation equivariance restricts the kernel to relative positions, $k(\mathbf{x}_j - \mathbf{x}_i)$. To achieve SE(3) equivariance on 3D data, the main approaches are *regular* and *steerable* group convolutions [Weiler et al., 2021, Brandstetter et al., 2022]. Steerable methods (a.k.a. tensor field networks [Thomas et al., 2018]) embed features in *group representations* (definition in App. A), which restricts non-linear operations to specialized equivariant ones to preserve these properties [Weiler and Cesa, 2019]. Regular group convolutions, conversely, lift features to an extended domain (e.g., $\mathbb{R}^3 \times \mathrm{SO}(3)$) to explicitly track group information, thereby permitting standard pointwise non-linearities [Cohen and Welling, 2016]. As our study isolates linear layer effects by fixing conventional non-linear components, we focus exclusively on *regular* group convolutions.

Within this regular group convolution paradigm, operations can be defined over the full group, creating feature fields on $\mathbb{R}^3 \times \mathrm{SO}(3)$ with convolutions of the form $[Lf](\mathbf{x}, \mathbf{R}) = \iint k(\mathbf{R}^T(\mathbf{x}' - \mathbf{x}), \mathbf{R}^T \mathbf{R}') f(\mathbf{x}', \mathbf{R}') d\mathbf{x}' d\mathbf{R}'$. For improved efficiency, these can be formulated over homogeneous spaces (definition in App. A) of the group. Our work adopts this strategy, utilizing feature fields over the position-orientation space $\mathbb{R}^3 \times S^2$. Convolutions are then given by $[Lf](\mathbf{x}, \mathbf{n}) = \iint k(\mathbf{R_n}^T(\mathbf{x}' - \mathbf{x}), \mathbf{R_n}^T \mathbf{n}') f(\mathbf{x}', \mathbf{n}') d\mathbf{x}' d\mathbf{n}'$, with $\mathbf{R_n}$ aligning a canonical axis to $\mathbf{n} \in S^2$, and reduce the integration domain from SO(3) to $S^2$ at the cost of moderate kernel symmetry constraints (e.g., axial symmetry [Bekkers et al., 2024]). A further simplification for SE(3) equivariance involves convolutions directly on $\mathbb{R}^3$ using isotropic kernels dependent only on distances, $k(\|\mathbf{x}_j - \mathbf{x}_i\|)$. This approach eliminates the orientation axis entirely, but imposes a more restrictive, non-directional kernel constraint.

## 3 Methodology

### 3.1 Expressivity and Equivariance Constraint Analysis

Our analysis of expressivity and equivariance constraints centers on our model, `Rapidash` (detailed in App. B). `Rapidash` employs regular group convolutions by processing feature fields $f : \mathbb{R}^3 \times S^2 \to \mathbb{R}^C$ over the position-orientation space. This involves convolution kernels on $\mathbb{R}^3 \times S^2$ that respect the quotient space symmetries ($\mathbb{R}^3 \times S^2 \equiv \mathrm{SE}(3)/\mathrm{SO}(2)$), a condition guaranteed to be met by conditioning message passing layers on the geometric invariants derived in [Bekkers et al., 2024].

A key theoretical aspect of `Rapidash`'s architecture is its fundamental connection to steerable tensor field networks. This relationship is established through Fourier analysis on the sphere $S^2$, where scalar fields can be decomposed into spherical harmonic coefficients that correspond to features transforming under irreducible representations of SO(3)—the building blocks of steerable networks. This equivalence is formally stated as:

**Proposition 3.1** (Equivalence via Fourier Transform). *A regular group convolution operating on scalar fields $f : \mathbb{R}^n \times Y \to \mathbb{R}^C$ (where $Y$ is $S^{n-1}$ or $SO(n)$) can be equivalently implemented as a steerable group convolution operating on fields of Fourier coefficients $\hat{f} : \mathbb{R}^n \to V_\rho$ (where $V_\rho$ is the space of combined irreducible representations) by performing point-wise Fourier transforms $\mathcal{F}_Y$ before the steerable convolution and inverse Fourier transforms $\mathcal{F}_Y^{-1}$ after.*

**Remark 1.** *This equivalence (discussed further by Bekkers et al. [2024, Appx. A.1], Brandstetter et al. [2022], and Cesa et al. [2021]) implies that* `Rapidash` *could, in principle, be reformulated as a tensor field network. Thus, our architecture inherently represents the capabilities of this widely used class of steerable networks.*

This connection to steerable networks also underpins its crucial universal approximation property. Building on established results for steerable GNNs [Dym and Maron, 2020] and related equivariant message passing schemes [Villar et al., 2021, Gasteiger et al., 2021], the SE(3)-equivariant universal approximation for `Rapidash` is formally stated as:

**Proposition 3.2** (Universal Approximation for Rapidash). `Rapidash`*, as an instance of message passing networks over* $\mathbb{R}^3 \times S^2$ *with message functions conditioned on the bijective invariant attributes (derived in [Bekkers et al., 2024, Thm. 1]), is an* SE(3)*-equivariant universal approximator. This specific universality follows from [Bekkers et al., 2024, Cor. 1.1], leveraging the sufficient expressivity of feature maps over* $\mathbb{R}^3 \times S^2$.

Regardless of `Rapidash`' strong theoretical expressivity, practical performance must weigh computational cost against actual expressivity. G-convs on extended domains like $\mathbb{R}^3 \times S^2$ involve feature fields of size, e.g., $P \times O \times C$ per point cloud (Points $\times$ Orientations $\times$ Channels), compared to $P \times C$ for $\mathbb{R}^3$-based models. While matching total features (e.g., $O \times C$ in an $\mathbb{R}^3 \times S^2$ model to an enhanced $C'$ in an $\mathbb{R}^3$ model) might suggest a nominally similar capacity, the $O$-axis in the former signifies a structured domain where features are correlated, not merely independent channels. The true expressive power, related to a model's hypothesis space size [Elesedy and Zaidi, 2021], varies with the imposed equivariance constraints: $T(3)$-equivariant $\mathbb{R}^3$ convolutions are least constrained, followed by SE(3)-equivariant $\mathbb{R}^3 \times S^2$ convolutions, and isotropic SE(3)-equivariant $\mathbb{R}^3$ convolutions being most constrained. For fair architectural comparison, we thus must consider scenarios where channel capacity (C) is maximized within practical limits. This leads to key research questions:

**Research Question 1: Kernel Constraint and Expressivity Impact** *How do different equivariance strategies—ordered from most to least constrained as previously outlined—impact task performance.*

**Research Question 2: Capacity Scaling and Generalization Gaps** *If more constrained SE(3)-equivariant models outperform $T(3)$-models (despite $T(3)$'s larger theoretical hypothesis space), can scaling $T(3)$-models (e.g., via channel capacity $C$ or training duration) close this apparent generalization gap? How do these architectures compare under nominally matched capacities?*

**Research Question 3: Data Scaling and Symmetry Relevance in Varied Contexts** *How does dataset size scaling impact the performance relativities of SE(3) versus $T(3)$-constrained models on SE(3)-symmetric tasks? In other contexts, such as tasks without SE(3) symmetry or low-data regimes, what is the practical generalization impact (benefit or hindrance) of imposing SE(3) constraints versus $T(3)$ constraints?*

### 3.2 Symmetry Breaking, and Generalization

This section analyzes generalization performance through the lens of probabilistic symmetry breaking, as introduced by Lawrence et al. [2025b] in their SymPE framework. Their work shows that incorporating informative auxiliary random variables $Z$ (such as pose information $Z \in SO(3)$) into a stochastic model $f(X, Z)$—that e.g. takes a featurized point cloud $X$ as input—can positively impact predictive inference, particularly when using jointly equivariant architectures. The ability of a model to leverage such an auxiliary variable $Z$ is reflected in the effective conditional distribution $\mathbb{P}(Z|X)$ that the model implicitly defines. We adopt this perspective to interpret how a model's architectural constraints, like those in our `Rapidash` architecture, affect its ability to use global orientation information, satisfying joint invariance $f(gX, gZ) = f(X, Z)$ for $g \in SO(3)$.

We analyze model expressivity by considering two scenarios for the conditional distribution $\mathbb{P}(Z|X)$ of the auxiliary pose $Z \in SO(3)$. First, if the pose $R_{\text{known}}$ is explicitly available (e.g., for aligned datasets like ShapeNet), $\mathbb{P}(Z|X) = \delta(Z - R_{\text{known}})$. This provides zero-entropy pose information, enabling a model $f(X, R_{\text{known}})$ to directly leverage this specific orientation. Second, standard $SO(3)$-*invariant* models $f_{\text{inv}}(X)$, which do not receive explicit pose input, operate as if $Z$ is drawn from a maximum entropy (uniform) distribution. This implies no specific orientation information is utilized, as derived in Proposition D.1. The disparity in available pose information—and thus in the effective entropy of $Z$—between these scenarios directly dictates model expressivity:

**Corollary 3.1** (Expressivity Gain from Low-Entropy Pose Information). *Let the optimal invariant mapping for a task, $f^*(X, R)$, depend non-trivially on canonical orientation $R$. Based on the distinct informational content of $Z$ outlined above: (a) Standard invariant models $f_{inv}(X)$ (maximum entropy pose) lack the expressivity to represent $f^*$; and (b) Pose-conditioned models $f_{cond}(X, R)$ (zero-entropy pose) can represent $f^*$.*

The enhanced expressivity afforded by conditioning on pose $R$, as established in Corollary 3.1, is fundamental for enabling symmetry breaking. By providing a determinate canonical reference $R$, models can disambiguate features arising from exact or approximate input symmetries, thereby producing more specific and potentially less symmetric outputs than standard equivariant models. We elaborate this viewpoint in context of ShapeNet segmentation in App. D.3 and generative modeling in App. D.4. Crucially, the well-known generalization advantages of equivariant architectures [Elesedy and Zaidi, 2021] also extend to models $f(\mathcal{X}, Z)$ that incorporate an auxiliary variable $Z$ for symmetry breaking, as formalized by Lawrence et al. [2025b, Thm. 6.1]. This general theorem confirms that structuring models $f(\mathcal{X}, Z)$ to be jointly equivariant is critical for realizing provable generalization gains when an auxiliary variable $Z$ is introduced (e.g., for symmetry breaking). Our Proposition D.2 details the application of this principle to our setting where $Z$ is a known, determinate pose $R$.

**Research Question 4: Empirical Benefits of Pose-Informed Symmetry Breaking** *Does incorporating explicit geometric information, such as pose $R$, to facilitate symmetry breaking lead to improved empirical performance compared to models that do not utilize such information?*

## 3.3 Equivariance breaking

Our architecture, `Rapidash`, is designed with the flexibility to process both scalar and vector-valued features at its input and output stages. For instance, it can map input vectors $v_i^{in}$ (e.g., initial velocities or normals) to spherical signals and predict output vectors $v_i^{out}$ (e.g., displacements) by appropriately projecting from spherical representations (see App. B for architectural specifics). This capability allows `Rapidash` to naturally incorporate informative geometric inputs such as global pose, normal vectors, or velocities. If these inputs are treated as proper geometric objects that transform consistently under $SE(3)$ actions, their inclusion can enhance the model's contextual understanding while preserving its overall $SE(3)$-equivariance.

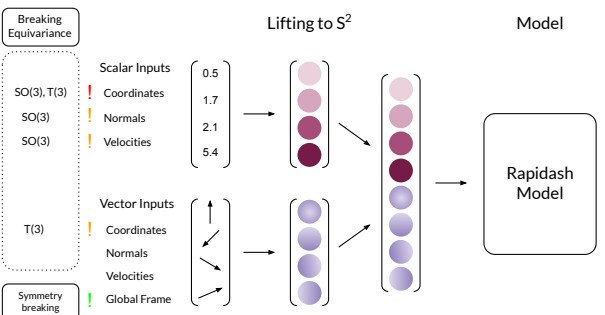

Figure 3: Input variations to Rapidash with base space $\mathbb{R}^3 \times S^2$ and $SE(3)$-equivariant convolutions lifted to $S^2$. It shows equivariance breaking through different inputs and symmetry breaking by using a global frame input.

This same flexibility in handling inputs also allows for controlled deviations from strict $SE(3)$-equivariance within `Rapidash`. For example, global coordinates or other geometric data can be supplied as fixed scalar features, which do not transform canonically under $SE(3)$ actions as true geometric vectors would (Fig.3). This presents a critical trade-off: while more input data can be powerful, what is the impact if it compromises the $SE(3)$-equivariance prior? As discussed in our related work (Section 4), the approximate and relaxed group equivariance approach suggests that such controlled deviations from strict symmetry can be beneficial, potentially improving training dynamics and performance. We explore this empirically with `Rapidash`, with the mechanisms for deviating from strict equivariance detailed in Appendix C. This leads to the following research question:

**Research Question 5: Equivariance-breaking** *What is the impact of breaking equivariance? Does providing more information to the model at the cost of breaking equivariance improve performance?*

## 4 Related Work

Weight sharing in convolutional networks was introduced in [LeCun et al., 2010] for images, and extended to group transformation in [Cohen and Welling, 2016] (recently formalized for MPNNs [Bekkers et al., 2024]). In the spirit of 'convolution is all you need' [Cohen et al., 2019], several works emerged like [Ravanbakhsh et al., 2017, Worrall et al., 2017, Kondor and Trivedi, 2018, Bekkers et al., 2018, Weiler et al., 2018, Cohen et al., 2019, Weiler and Cesa, 2019, Bekkers, 2019, Sosnovik et al., 2019, Finzi et al., 2020] presenting different frameworks for group convolutional architectures.

Wang et al. [2022], van der Ouderaa et al. [2022], Kim et al. [2023], Pertigkiozoglou et al. [2024] demonstrate that controlled equivariance breaking can significantly improve performance, also highlighting the benefits of relaxed group equivariance constraints. Similarly, work by Petrache and Trivedi [2023] suggests that while aligning data symmetry with architectural symmetry (strict equivariance) is ideal, models with partial or approximate equivariance can still exhibit improved generalization compared to fully non-equivariant ones. More recent works show different ways to break symmetry, spontaneous symmetry breaking in [Xie and Smidt, 2024] and symmetry-breaking via random canonicalization in [Lawrence et al., 2025b]. For additional details on internal vs external symmetry and the relation of existing works, see App. C.

The literature presents conflicting evidence on the practical benefits of strict equivariance versus less constrained models. For example, while equivariant models with higher-order tensor representations (e.g., E3GNN, NequIP [Batzner et al., 2022], SEGNN [Brandstetter et al., 2022]) have shown improved data efficiency and performance for tasks like molecular interatomic potentials, findings in other domains differ. Thais and Murnane [2023] reported that Lorentz equivariant models [Brehmer et al., 2023] offered no clear advantage over non-equivariant counterparts in particle physics. Similarly, in materials science, though $SO(3)$-equivariant Graph Transformers [Liao and Smidt, 2023, Liao et al., 2024] achieved strong results, Qu and Krishnapriyan [2024] (EScAIP) demonstrated that extensively scaled non-equivariant models can be competitive or even superior. Conversely, scaling experiments by Brehmer et al. [2024] using transformers for rigid-body simulations argued in favor of equivariant networks. Such divergent outcomes highlight the challenge of drawing universal conclusions due to common variations in model architectures and datasets, a difficulty our work aims to address by introducing `Rapidash` as a unified framework for controlled comparative studies.

## 5 Experiments and Results

In each table, the top section shows models with $SE(3)$ equivariance, and the bottom with $T_3$ equivariance. Equivariance breaking due to the input specification is indicated by: ❗ (breaking $SE(3)$), ❗ (break $T_3$ or $SO(3)$), while ❗ denotes that the model is no longer equivariant. All models for equivariant tasks are trained with $SO(3)$ augmentation. A ✓ indicates an option is used, ✗, indicates it is not used, and "-" indicates the option is not available. Lastly, we add references from literature with state-of-the-art (SotA) performance. We emphasize that our experiments aim to demonstrate how a group convolutional architecture and its variants perform to answer the proposed Research Questions, instead of having results that surpass SotA models, but we nevertheless find it important

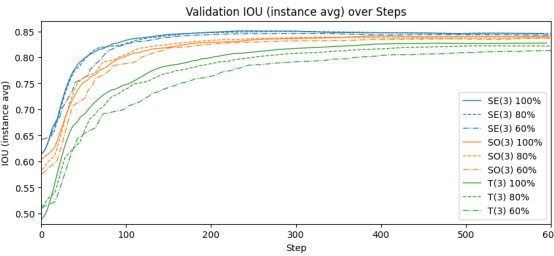

Figure 4: Smoothed IoU (Instance average) performance curves over time for ShapeNet part segmentation. The legend codes correspond to the effective equivariance and the percentage of the training dataset trained on.

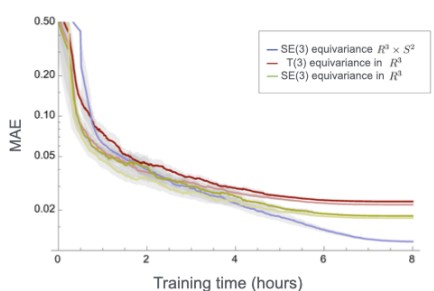

Figure 5: MAE ($10^{-3}$) vs training time for QM9 property ($\mu$) prediction task.

to provide references from literature as they indicate our results are close to, or surpass, the state of the art, which validates the correctness of our implementation and thus the validity of our results. We list the most notable results in the tables of the main text while presenting full extended versions in the Appendix.

**Molecular property prediction & generation task**   For predicting molecular properties and generating molecules, we use QM9 [Ramakrishnan R., 2014]. We evaluate the prediction of molecular properties using the MAE and compare these with EGNN [Satorras et al., 2021], Dimenet++ [Gasteiger et al., 2022], and SE(3)- Transformer [Fuchs et al., 2020]. For the molecule generation, we train a generative model that uses equivariant denoising layers F. We evaluate molecular generation on metrics from Hoogeboom et al. [2022] that include atomic stability, molecule stability, as well as a new aggregate metric *discovery*, which is a fraction of generated samples that are jointly valid, unique, and new. We compare our models with denoising diffusion [Ho et al., 2020] model like EDM [Hoogeboom et al., 2022], PΘNITA [Bekkers et al., 2024], and MuDiff [Hua et al., 2024].

Table 1: Ablation results on QM9 for property prediction and generation. For prediction, we report mean absolute error (MAE) ($\times 10^{-3}$). For generation, we evaluate atom stability, molecule stability, and *discovery*.

| | | ! Coordinates as Scalars | ! Coordinates as Vectors | | Regression | | | Generation | | | |
|---|---|---|---|---|---|---|---|---|---|---|---|
| Model Variation | Type | | | Effective Equivariance | MAE $\mu$ (D) | MAE $\alpha$ ($a_o^3$) | MAE $\epsilon_{HOMO}$ (eV) | Stab Atom % | Stab Mol % | Discover % | Normalized Epoch Time |
| colspan | | | | | *Rapidash with internal SE(3) Equivariance Constraint* | | | | | | |
| 1 | $\mathbb{R}^3$ | ✗ | - | SE(3) | **17.41**$_{\pm0.37}$ | 53.03$_{\pm0.92}$ | **22.29**$_{\pm0.27}$ | - | - | - | ~1 |
| 2 | $\mathbb{R}^3$ | ✓ | - | none | 17.77$_{\pm0.21}$ | **52.22**$_{\pm0.93}$ | 22.44$_{\pm0.15}$ | **96.74**$_{\pm0.07}$ | **72.39**$_{\pm0.50}$ | **89.06**$_{\pm0.39}$ | ~1 |
| 3 | $\mathbb{R}^3 \times S^2$ | ✗ | ✗ | SE(3) | **10.39**$_{\pm0.33}$ | 42.20$_{\pm1.17}$ | 19.30$_{\pm1.02}$ | **99.38**$_{\pm0.01}$ | **93.12**$_{\pm0.28}$ | 90.78$_{\pm0.11}$ | ~10 |
| 4 | $\mathbb{R}^3 \times S^2$ | ✗ | ✓ | SO(3) | 10.44$_{\pm0.22}$ | 42.68$_{\pm1.04}$ | 19.64$_{\pm0.75}$ | **99.38**$_{\pm0.02}$ | 92.91$_{\pm0.41}$ | 90.26$_{\pm0.20}$ | ~10 |
| 5 | $\mathbb{R}^3 \times S^2$ | ✓ | ✗ | none | 10.53$_{\pm0.17}$ | **40.21**$_{\pm0.56}$ | **18.69**$_{\pm0.20}$ | 99.33$_{\pm0.06}$ | 92.71$_{\pm0.51}$ | **91.54**$_{\pm0.34}$ | ~10 |
| colspan | | | | | *Rapidash with Internal $T_3$ Equivariance Constraint* | | | | | | |
| 6 | $\mathbb{R}^3$ | ✗ | - | $T_3$ | 22.11$_{\pm1.03}$ | 61.97$_{\pm1.05}$ | 25.83$_{\pm0.25}$ | **98.57**$_{\pm0.01}$ | **81.62**$_{\pm0.15}$ | **91.83**$_{\pm0.45}$ | ~1 |
| 7 | $\mathbb{R}^3$ | ✓ | - | none | **21.40**$_{\pm0.37}$ | **58.18**$_{\pm0.09}$ | **24.54**$_{\pm0.26}$ | 98.40$_{\pm0.02}$ | 78.48$_{\pm0.36}$ | 89.26$_{\pm0.30}$ | ~1 |
| colspan | | | | | *Reference methods from literature* | | | | | | |
| EGNN | - | - | - | - | 29.0 | 71.0 | 29.0 | 98.7 | 82.0 | - | - |
| DimeNet++ | - | - | - | - | 29.7 | 43.5 | 24.6 | - | - | - | - |
| SE(3)-T | - | - | - | - | 53.0 | 51.0 | 53.0 | - | - | - | - |
| MuDiff | - | - | - | - | - | - | - | 98.8 | 89.9 | - | - |
| EGNN-EDM | - | - | - | - | - | - | - | 99.3 | 90.7 | 89.5 | - |
| PΘNITA | - | - | - | - | - | - | - | 98.9 | 87.8 | - | - |

Tab. 1 summarizes results on QM9. For *property prediction*, SE(3)-equivariant Rapidash variants consistently outperform their $T_3$-equivariant counterparts, and the performance gap does not close with extended training (Fig.5). Among these SE(3) models, those utilizing the $\mathbb{R}^3 \times \mathbb{S}^2$ base type achieve superior accuracy over $\mathbb{R}^3$-only versions—highlighting the benefits of directional input and internal directional representations—and notably reach performance levels on par with established state-of-the-art methods on several regression tasks. We also find that breaking equivariance by adding scalar coordinate features further benefits both base types, likely by helping to resolve symmetric ambiguities. In *molecule generation*, the best results are decisively achieved by our SE(3)-equivariant Rapidash models, again favoring the $\mathbb{R}^3 \times \mathbb{S}^2$ base type, which demonstrate strong molecular stability in contrast to $T_3$ variants. Remarkably, on this generative task, these models significantly surpass existing state-of-the-art performance. Collectively, these outcomes demonstrate that Rapidash's core convolutional architecture, while designed for flexibility and controlled experimentation across its variants, is itself highly effective and achieves competitive to leading results in both predictive and generative domains.

**3D point cloud segmentation & generation task**   We evaluate our models on ShapeNet3D [Chang et al., 2015] for part segmentation and generation tasks. We additionally compare our models to various SotA methods like PointNeXt [Qian et al., 2022], Deltaconv [Wiersma et al., 2022], and GeomGCNN [Srivastava and Sharma, 2021] for the part segmentation task. For the generation task, we compare to LION, [Zeng et al., 2022], PVD Zhou et al. [2021], and DPM Luo and Hu [2021].

Tab. 2 shows our results on the ShapeNet dataset. For *segmentation*, especially on out-of-distribution (rotated) inputs, SE(3)-equivariant Rapidash variants over $\mathbb{R}^3 \times \mathbb{S}^2$ consistently perform best, echoing our QM9 findings. Performance is further enhanced by explicit symmetry breaking (e.g., adding a global frame for a stable orientation reference) or by specific types of equivariance breaking,

Table 2: Ablation results on ShapeNet3D and CMU datasets. For ShapeNet3D, we report part segmentation performance using mean Intersection over Union (mIoU) on aligned and randomly rotated inputs, and 1-Nearest Neighbor Accuracy (1-NNA) for shape generation. For CMU motion prediction, we report mean squared error (MSE) scaled by $10^{-2}$.

| Model Variation | Type | Coordinates as Scalars ! | Coordinates as Vectors ! | Normals\Velocity as Scalars ! | Normals\Velocity as Vectors ! | Symmetry Breaking ! | Effective Equivariance | ShapeNet IoU↑ (aligned) | IoU↑ (rotated) | 1-NNA↓ CD | CMU MSE↓ | MSE↓ (rotated) |
|---|---|---|---|---|---|---|---|---|---|---|---|---|
| | | | | | | | | **ShapeNet** | | | **CMU** | |
| colspan Rapidash with internal SE(3) Equivariance Constraint |
| 1 | $\mathbb{R}^3$ | ✗ | - | ✗ | - | - | SE(3) | $79.08_{\pm0.22}$ | $\mathbf{79.08}_{\pm0.33}$ | - | $>100$ | $>100$ |
| 2 | $\mathbb{R}^3$ | ✗ | - | ✓ | - | - | $T_3$ | $83.54_{\pm0.04}$ | $49.06_{\pm2.27}$ | - | $\mathbf{4.95}_{\pm0.11}$ | $24.62_{\pm2.83}$ |
| 3 | $\mathbb{R}^3$ | ✓ | - | ✗ | - | - | none | $83.65_{\pm0.03}$ | $32.14_{\pm0.17}$ | $\mathbf{69.19}_{\pm0.30}$ | $6.77_{\pm0.06}$ | $92.10_{\pm9.41}$ |
| 4 | $\mathbb{R}^3$ | ✓ | - | ✓ | - | - | none | $\mathbf{84.43}_{\pm0.11}$ | $33.83_{\pm0.36}$ | - | $6.04_{\pm0.21}$ | $>100$ |
| 5 | $\mathbb{R}^3\times S^2$ | ✗ | ✗ | ✗ | ✗ | ✗ | SE(3) | $84.27_{\pm0.19}$ | $84.17_{\pm0.08}$ | - | $5.43_{\pm0.15}$ | $5.42_{\pm0.16}$ |
| 6 | $\mathbb{R}^3\times S^2$ | ✗ | ✗ | ✗ | ✗ | ✓ | SE(3) | $85.44_{\pm0.03}$ | $85.39_{\pm0.11}$ | $65.16_{\pm0.47}$ | $5.73_{\pm0.14}$ | $5.73_{\pm0.13}$ |
| 7 | $\mathbb{R}^3\times S^2$ | ✗ | ✗ | ✗ | ✓ | ✗ | SE(3) | $84.75_{\pm0.19}$ | $84.65_{\pm0.20}$ | - | $4.77_{\pm0.11}$ | $4.78_{\pm0.12}$ |
| 8 | $\mathbb{R}^3\times S^2$ | ✗ | ✗ | ✗ | ✓ | ✓ | SE(3) | $85.48_{\pm0.14}$ | $85.41_{\pm0.09}$ | - | $\mathbf{4.69}_{\pm0.10}$ | $\mathbf{4.70}_{\pm0.10}$ |
| 9 | $\mathbb{R}^3\times S^2$ | ✗ | ✓ | ✗ | ✗ | ✗ | SO(3) | $84.47_{\pm0.15}$ | $84.56_{\pm0.02}$ | - | $8.07_{\pm0.30}$ | $8.05_{\pm0.28}$ |
| 10 | $\mathbb{R}^3\times S^2$ | ✗ | ✓ | ✗ | ✗ | ✓ | SO(3) | $85.38_{\pm0.04}$ | $85.36_{\pm0.05}$ | $\mathbf{62.01}_{\pm1.24}$ | $5.36_{\pm0.14}$ | $5.37_{\pm0.13}$ |
| 11 | $\mathbb{R}^3\times S^2$ | ✗ | ✓ | ✗ | ✓ | ✗ | SO(3) | $84.79_{\pm0.16}$ | $84.67_{\pm0.21}$ | - | $5.21_{\pm0.18}$ | $5.22_{\pm0.17}$ |
| 12 | $\mathbb{R}^3\times S^2$ | ✗ | ✓ | ✗ | ✓ | ✓ | SO(3) | $\mathbf{85.76}_{\pm0.04}$ | $\mathbf{85.74}_{\pm0.05}$ | - | $4.88_{\pm0.04}$ | $4.89_{\pm0.04}$ |
| 13 | $\mathbb{R}^3\times S^2$ | ✗ | ✗ | ✓ | ✗ | ✗ | $T_3$ | $85.46_{\pm0.14}$ | $42.93_{\pm2.58}$ | - | $4.73_{\pm0.08}$ | $32.47_{\pm1.97}$ |
| 14 | $\mathbb{R}^3\times S^2$ | ✗ | ✗ | ✗ | ✗ | ✗ | none | $85.34_{\pm0.17}$ | $36.65_{\pm2.52}$ | $\mathbf{61.87}_{\pm0.36}$ | $5.37_{\pm0.06}$ | $46.69_{\pm0.27}$ |
| 15 | $\mathbb{R}^3\times S^2$ | ✓ | ✗ | ✓ | ✗ | ✗ | none | $84.93_{\pm0.36}$ | $33.35_{\pm1.66}$ | - | $5.80_{\pm0.20}$ | $58.66_{\pm1.72}$ |
| colspan Rapidash with Internal $T_3$ Equivariance Constraint |
| 16 | $\mathbb{R}^3$ | ✗ | - | ✗ | - | - | $T_3$ | $85.20_{\pm0.15}$ | $31.58_{\pm0.32}$ | $65.81_{\pm0.26}$ | $5.56_{\pm0.05}$ | $52.68_{\pm1.26}$ |
| 17 | $\mathbb{R}^3$ | ✗ | - | ✓ | - | - | $T_3$ | $85.24_{\pm0.07}$ | $\mathbf{33.11}_{\pm0.81}$ | - | $\mathbf{5.21}_{\pm0.18}$ | $\mathbf{39.15}_{\pm0.69}$ |
| 18 | $\mathbb{R}^3$ | ✓ | - | ✗ | - | - | none | $\mathbf{85.26}_{\pm0.08}$ | $31.10_{\pm0.77}$ | $\mathbf{62.29}_{\pm0.43}$ | $6.24_{\pm0.09}$ | $77.98_{\pm6.39}$ |
| 19 | $\mathbb{R}^3$ | ✓ | - | ✓ | - | - | none | $84.29_{\pm0.17}$ | $32.97_{\pm0.59}$ | - | $5.99_{\pm0.23}$ | $>100$ |
| colspan Reference methods from literature (ShapeNet 3D) |
| LION | - | - | - | - | - | - | - | - | - | 51.85 | - | |
| PVD | - | - | - | - | - | - | - | - | - | 58.65 | - | |
| DPM | - | - | - | - | - | - | - | - | - | 62.30 | - | |
| DeltaConv | - | - | - | - | - | - | - | 86.90 | - | - | - | |
| PointNeXt | - | - | - | - | - | - | - | 87.00 | - | - | - | |
| GeomGCNN * | - | - | - | - | - | - | - | 89.10 | - | - | - | |
| colspan Reference methods from literature (CMU Motion) |
| TFN | - | - | - | - | - | - | - | - | - | - | 66.90 | - |
| EGNN | - | - | - | - | - | - | - | - | - | - | 31.70 | - |
| CGENN | - | - | - | - | - | - | - | - | - | - | 9.41 | - |
| CSMPN ** | - | - | - | - | - | - | - | - | - | - | 7.55 | - |

\* is trained with 1024 points, instead of 2048. \*\* uses additional pre-computed simplicial structures in training data.

such as treating input coordinates as fixed vector features to encode absolute spatial location, as may be expected for this non-symmetry-critical task. For 3D shape *generation*, which aims to produce *canonically aligned objects* and thus neither require full SE(3)-equivariance, models based on the SE(3)-equivariant $\mathbb{R}^3 \times \mathbb{S}^2$ architecture remain highly effective. Notably, $\mathbb{R}^3 \times \mathbb{S}^2$ models that incorporate mechanisms that break equivariance achieve the top generative results. The pose-informed, symmetry-broken variants even slightly outperform the translation-equivariant $\mathbb{R}^3$ models.

**Human motion prediction task** We next evaluate the model variations on the CMU Human Motion Capture dataset [Gross and Shi, 2001](200 training samples), where the task is to predict motion trajectory (Fig.11). We list NRI [Kipf et al., 2018], EGNN [Satorras et al., 2021], CEGNN [Ruhe et al., 2023] and CSMPN [Liu et al., 2024] models from literature.

Tab. 2 reports results on the CMU motion capture dataset. We observe that SE(3)-equivariant models over $\mathbb{R}^3 \times \mathbb{S}^2$ with velocity vector inputs achieve the lowest MSE. Note that such models fully equivariantly map from input to output (a vector). Breaking symmetry by means of a global frame further improves performance in most cases. In contrast, models operating solely over $\mathbb{R}^3$ perform poorly, particularly under (out of distribution) rotations, highlighting the importance of a capability to equivariantly handle directional information. Recall that the SE(3) equivariant $\mathbb{R}^3$ model 1 cannot solve this task, as it is a strictly invariant model, and the task requires predicting a non-trivially transforming velocity vector.

**Representation capacity and data efficiency** To study the effect of representation capacity, we scale up the number of channels in the $\mathbb{R}^3$ models to try to close a potential performance gap to the $\mathbb{R}^3 \times \mathbb{S}^2$ due to sub-optimal network capacity. As shown in Tab. 3 and App. Tab. 5, this increase has little impact on performance, suggesting that all model variants are already operating near their effective capacity limits. As an additional experiment, we also evaluate data efficiency on ShapeNet (Fig. 4) and find that models with stronger equivariance require fewer training samples to converge, indicating the advantage of equivariance in low-data settings.

Table 3: Comparing models with inflated representation capacity against regular models on ShapeNet3D for the part segmentation task for aligned and randomly rotated samples.

| Model Variation | Type | Coordinates as Scalars ! | Coordinates as Vectors ! | Normals as Scalars ! | Normals as Vectors | Symmetry Breaking ! | Effective Equivariance | IoU↑ (aligned) | IoU↑ (rotated) | Normalized Epoch Time |
|---|---|---|---|---|---|---|---|---|---|---|
| \multicolumn{11}{c}{Rapidash with internal SE(3) Equivariance Constraint} |
| 1 | $\mathbb{R}^3$ | ✗ | - | ✗ | - | - | SE(3) | $80.26_{\pm 0.06}$ $80.31_{\pm 0.15}$ | $\mathbf{80.33}_{\pm 0.13}$ $80.20_{\pm 0.07}$ | $\sim 1$ $\sim 10$ |
| 2 | $\mathbb{R}^3$ | ✗ | - | ✓ | - | - | $T_3$ | $83.95_{\pm 0.06}$ $83.87_{\pm 0.09}$ | $52.09_{\pm 0.87}$ $49.63_{\pm 0.87}$ | $\sim 1$ $\sim 10$ |
| 3 | $\mathbb{R}^3$ | ✓ | - | ✗ | - | - | none | $84.23_{\pm 0.08}$ $84.01_{\pm 0.06}$ | $34.15_{\pm 0.05}$ $32.22_{\pm 0.27}$ | $\sim 1$ $\sim 10$ |
| 4 | $\mathbb{R}^3$ | ✓ | - | ✓ | - | - | none | $\mathbf{84.75}_{\pm 0.02}$ $84.48_{\pm 0.16}$ | $34.07_{\pm 0.43}$ $32.90_{\pm 0.47}$ | $\sim 1$ $\sim 10$ |
| \multicolumn{11}{c}{Rapidash with Internal $T_3$ Equivariance Constraint} |
| 16 | $\mathbb{R}^3$ | ✗ | - | ✗ | - | - | $T_3$ | $85.26_{\pm 0.01}$ $85.38_{\pm 0.05}$ | $31.41_{\pm 0.86}$ $32.78_{\pm 0.59}$ | $\sim 1$ $\sim 10$ |
| 17 | $\mathbb{R}^3$ | ✗ | - | ✓ | - | - | $T_3$ | $\mathbf{85.82}_{\pm 0.10}$ $85.71_{\pm 0.17}$ | $\mathbf{35.42}_{\pm 0.98}$ $32.25_{\pm 0.07}$ | $\sim 1$ $\sim 10$ |
| 18 | $\mathbb{R}^3$ | ✓ | - | ✗ | - | - | none | $85.51_{\pm 0.06}$ $85.52_{\pm 0.09}$ | $32.79_{\pm 0.74}$ $31.11_{\pm 0.15}$ | $\sim 1$ $\sim 10$ |
| 19 | $\mathbb{R}^3$ | ✓ | - | ✓ | - | - | none | $85.66_{\pm 0.02}$ $85.16_{\pm 0.06}$ | $33.55_{\pm 0.49}$ $30.62_{\pm 0.21}$ | $\sim 1$ $\sim 10$ |

## 6 Discussion

Regarding the **impact of kernel constraints (RQ1)**, our findings indicate that performance is dictated more by the *appropriateness and geometric expressivity of the inductive bias* than by merely minimizing group constraints, under optimized model capacities. For instance, on aligned ShapeNet data (Table 2), where SO(3)-equivariance demands might seem harsh, the less constrained $T_3$-equivariant $\mathbb{R}^3$ models perform comparably to, not better than, the richer SE(3)-equivariant $\mathbb{R}^3 \times S^2$ variants. Within SE(3) models, highly constrained isotropic $\mathbb{R}^3$ kernels consistently underperform against the less constrained $\mathbb{R}^3 \times \mathbb{S}^2$ kernels across tasks. Furthermore, on SE(3)-critical tasks (QM9 property prediction, Table 1; CMU motion, Table 2), the $\mathbb{R}^3 \times S^2$ models significantly surpass the $T_3$ $\mathbb{R}^3$ models. It underscores that a well-suited inductive bias is critical.

Regarding **capacity scaling and generalization gaps (RQ2)**, our findings indicate that while increased model capacity (e.g., "inflated" models in Fig. 1, Tab. 3 and App. Tab. 5) benefits both $T_3$ and SE(3) approaches, it *does not consistently allow* simpler $T_3$-based $\mathbb{R}^3$ models to *close the performance gap* to SE(3)-equivariant models on the richer $\mathbb{R}^3 \times S^2$ domain. Surprisingly, the latter often achieve superior results more efficiently within a given computational budget despite the higher computational complexity (Fig. 1). Moreover, performance frequently *saturates* with increasing capacity (Tab. 3), suggesting that models reach the *limits of their architecturally-defined hypothesis space* for geometric expressivity, rather than merely lacking parameters. Thus, the fundamental *architectural choice often outweighs sheer capacity* and computational resources in bridging these generalization gaps. This claim is further underpinned by additional scaling experiments on QM9, which show that the SE(3) inductive bias maintains its significant advantage even when $T_3$ models are trained with matched total representational capacity and comparable computational time (App. E.2).

The influence of **data scaling and symmetry relevance (RQ3)** highlights that SE(3)-equivariant models particularly excel in *low-data regimes*, showcasing strong data efficiency, as seen on the intrinsically small CMU motion dataset and with reduced ShapeNet training data (Fig. 4). Notably, this data-efficiency benefit of SE(3) group convolutions persists even for ShapeNet's aligned data, where strict SO(3)-equivariance for the specific output alignment might seem less critical.

Concerning **pose-informed symmetry breaking (RQ4)**, our experiments affirm its empirical benefits. Incorporating explicit geometric reference information, such as a global frame, consistently improved performance across diverse tasks like ShapeNet segmentation and CMU motion prediction (Tab. 2). This aligns with the theoretical expectation (Corollary 3.1, App. D.3, and App. D.4) that providing a canonical reference frame enhances expressivity by allowing models to disambiguate inputs.

Finally, regarding **equivariance-breaking by introducing additional information (RQ5)**, the impact is nuanced and task-dependent, as illustrated in Fig. 2. Providing geometric information in a manner that breaks strict SE(3)-equivariance (e.g., coordinates as scalar features) proved advantageous for QM9 tasks (Tab. 1) and for ShapeNet generation (Tab. 2), where it likely helped capture task-relevant

non-symmetric aspects and absolute spatial references. It also benefited ShapeNet segmentation on non-aligned samples. However, this is does not seem a universal strategy for improvement, as indiscriminate breaking or unsuitable information can be neutral or detrimental to performance.

**Limitations**  Our empirical explorations, while extensive and in line with typical academic resources (e.g., training for Fig. 1 was within a 9-hour budget per run, and tabular results based on up to 1000 epochs), used a modest computational budget. This could limit the exhaustive exploration of scaling laws (RQ2) for simpler $\mathbb{R}^3$-based models, although observed performance saturation suggests these specific architectures may have already approached their practical limits within our framework. Furthermore, this study's scope is centered on convolutional architectures applied to point cloud data; while we believe many principles discussed may generalize, an explicit investigation into other modalities or architectures, such as transformers, remains future work.

## 7  Conclusion

In this paper, we aimed to constructively contribute to an ongoing discussion surrounding the interplay of scale and equivariance in deep learning. We divide this broad discussion into concrete research questions (RQ1-5) and then address these through exhaustive experimentation and analysis. Through the introduction of `Rapidash`—a general regular group convolutional architecture enabling controlled comparisons, we have sought to provide empirical insights into these critical trade-offs with an extensive suite of nuanced experiments. We couple these results with targeted theoretical analysis of equivariance, symmetry breaking, and generalization.

Our findings offer a structured understanding of when and how different equivariance strategies and symmetry considerations benefit practical model performance, thereby offering guidance for future architectural design and model selection in this domain.

## Acknowledgements

Shavaree Vadgama is funded by the Hybrid Intelligence Center, a 10-year programme funded through the research programme Gravitation, which is (partly) financed by the Dutch Research Council (NWO). This publication is part of the project SIGN with file number VI.Vidi.233.220 of the research programme Vidi, which is (partly) financed by the Dutch Research Council (NWO) under the grant https://doi.org/10.61686/PKQGZ71565.

SV would like to express sincere appreciation for the valuable feedback provided by Tejaswi Kasarla and Riccardo Valperga. SV is very grateful for the detailed discussion on the manuscript with members of the LCV group at NYU/Flatiron Institute. The authors gratefully acknowledge the reviewers for their detailed and constructive feedback, which helped strengthen this work.

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

# A    Mathematical Prerequisites and Notations

**Groups.** A *group* is an algebraic structure defined by a set $G$ and a binary operator $\cdot : G \times G \to G$, known as the *group product*. This structure $(G, \cdot)$ must satisfy four axioms: (1) *closure*, where $\forall_{h,g \in G} : h \cdot g \in G$; (2) the existence of an *identity* element $e \in G$ such that $\forall_{g \in G}, e \cdot g = g \cdot e = g$, (3) the existence of an *inverse* element, i.e. $\forall_{g \in G}$ there exists a $g^{-1} \in G$ such that $g^{-1} \cdot g = e$; and (4) *associativity*, where $\forall_{g,h,p \in G} : (g \cdot h) \cdot p = g \cdot (h \cdot p)$. Going forward, group product between two elements will be denoted as $g, g' \in G$ by juxtaposition, i.e., as $g\, g'$.

For Special Euclidean group $\mathrm{SE}(n)$, the group product between two roto-translations $g=(\mathbf{x}, \mathbf{R})$ and $g'=(\mathbf{x}', \mathbf{R}')$ is given by $(\mathbf{x}, \mathbf{R})\,(\mathbf{x}', \mathbf{R}')=(\mathbf{R}\mathbf{x}' + \mathbf{x}, \mathbf{R}\,\mathbf{R}')$, and its identity element is given by $e=(\mathbf{0}, \mathbf{I})$.

**Homogeneous Spaces.** A group can act on spaces other than itself via a *group action* $gT : G \times X \to X$, where $X$ is the space on which $G$ acts. For simplicity, the action of $g \in G$ on $x \in X$ is denoted as $g\,x$. Such a transformation is called a group action if it is homomorphic to $G$ and its group product. That is, it follows the group structure: $(g\, g')\,x=g\,(g'\, x)\ \forall g, g' \in G, x \in X$, and $e\,x=x$. For example, consider the space of 3D positions $X = \mathbb{R}^3$, e.g., atomic coordinates, acted upon by the group $G=\mathrm{SE}(3)$. A position $\mathbf{p} \in \mathbb{R}^3$ is roto-translated by the action of an element $(\mathbf{x}, \mathbf{R}) \in SE(3)$ as $(\mathbf{x}, \mathbf{R})\,\mathbf{p}=\mathbf{R}\,\mathbf{p} + \mathbf{x}$.

A group action is termed *transitive* if every element $x \in X$ can be reached from an arbitrary origin $x_0 \in X$ through the action of some $g \in G$, i.e., $x=gx_0$. A space $X$ equipped with a transitive action of $G$ is called a *homogeneous space* of $G$. Finally, the *orbit* $G\,x := \{g\,x \mid g \in G\}$ of an element $x$ under the action of a group $G$ represents the set of all possible transformations of $x$ by $G$. For homogeneous spaces, $X=G\,x_0$ for any arbitrary origin $x_0 \in X$.

**Quotient spaces.** The aforementioned space of 3D positions $X=\mathbb{R}^3$ serves as a homogeneous space of $G = \mathrm{SE}(3)$, as every element $\mathbf{p}$ can be reached by a roto-translation from $\mathbf{0}$, i.e., for every $\mathbf{p}$ there exists a $(\mathbf{x}, \mathbf{R})$ such that $\mathbf{p}=(\mathbf{x}, \mathbf{R})\,\mathbf{0}=\mathbf{R}\,\mathbf{0} + \mathbf{x}=\mathbf{x}$. Note that there are several elements in $SE(3)$ that transport the origin $\mathbf{0}$ to $\mathbf{p}$, as any action with a translation vector $\mathbf{x}=\mathbf{p}$ suffices regardless of the rotation $\mathbf{R}$. This is because any rotation $\mathbf{R}' \in \mathrm{SO}(3)$ leaves the origin unaltered.

We denote the set of all elements in $G$ that leave an origin $x_0 \in X$ unaltered the *stabilizer subgroup* $\mathrm{Stab}_G(x_0)$. In subsequent analyses, the symbol $H$ is used to denote the stabilizer subgroup of a chosen origin $x_0$ in a homogeneous space, i.e., $H=\mathrm{Stab}_G(x_0)$. We further denote the *left coset* of $H$ in $G$ as $g\,H := \{g\,h \mid h \in H\}$. In the example of positions $\mathbf{p} \in X=\mathbb{R}^3$ we concluded that we can associate a point $\mathbf{p}$ with many group elements $g \in SE(3)$ that satisfy $\mathbf{p}=g\,\mathbf{0}$. In general, letting $g_x$ be any group element s.t. $x=g_x\,x_0$, then any group element in the left set $g_x\,H$ is also identified with the point $\mathbf{p}$. Hence, any $x \in X$ can be identified with a left coset $g_xH$ and vice versa.

Left cosets $g\,H$ then establish an *equivalence relation* $\sim$ among transformations in $G$. We say that two elements $g, g' \in G$ are equivalent, i.e., $g \sim g'$, if and only if $g\,x_0=g'\,x_0$. That is, if they belong to the same coset $g\,H$. The space of left cosets is commonly referred to as the *quotient space* $G/H$.

We consider *feature maps* $f : X \to \mathbb{R}^C$ as multi-channel signals over homogeneous spaces $X$. Here, we treat point clouds as sparse feature maps, e.g., sampled only at atomic positions. In the general continuous setting, we denote the space of feature maps over $X$ with $\mathcal{X}$. Such feature maps undergo group transformations through *regular group representations* $\rho^{\mathcal{X}}(g) : \mathcal{X} \to \mathcal{X}$ parameterized by $g$, and which transform functions $f \in \mathcal{X}$ via $[\rho^{\mathcal{X}}(g)f](x)=f(g^{-1}x)$ .

**Irreducible representations and spherical harmonics** Given any representation, there are often orthogonal subspaces that do not interact with each other, making it possible to break our representation down into smaller pieces by restricting to these subspaces. Hence, it is useful to consider the representations that cannot be broken down. Such representations are terms irreducible representations or irreps. Given a group $G$, $V$ a vector space, and $\rho : G \to GL(V)$ representation, a representation is irreducible if there is no nontrivial proper subspace $W \subset V$ such that $\rho|_W$ is a representation of $G$ over space $W$. With each irrep there is an associated (harmonic) frequency $l$. The irreps of $SO(3)$ are given by the $(2l + 1) \times (2l + 1)$ dimensional rotation matrices called *Wigner-D matrices*. The central columns of these matrices comprise the set of $2l + 1$ *spherical harmonics* $Y_m^{(l)} : \mathbb{S}^2 \to \mathbb{R}$, indexed by $m = -l, ..., l$.

# B   `Rapidash`: A Regular Group Convolution Approach for Flexible Equivariant Modeling

The `Rapidash` architecture, employed throughout our empirical study, builds upon the efficient SE(3)-equivariant regular group convolution framework operating on position-orientation space ($\mathbb{R}^3 \times S^2$) as introduced by PΘNITA [Bekkers et al., 2024]. To facilitate a comprehensive investigation into the utility of equivariance and symmetry breaking, `Rapidash` extends this foundation by incorporating several key flexibilities: (i) versatile handling of various input and output geometric quantities (e.g., scalars, vectors representing positions, normals, or pose information); (ii) enhanced scalability for large point clouds through multi-scale processing incorporating techniques like farthest point sampling; and (iii) convenient adaptability to different equivariance constraints, allowing for controlled comparisons between $SE(n)$ and translation-only ($T_n$) equivariant models. Like PΘNITA, `Rapidash` primarily adopts the regular group convolution paradigm, distinguishing it from steerable G-CNNs or tensor field networks, although fundamental connections exist. This section elucidates the theoretical underpinnings of this approach.

## B.1   Regular vs. Steerable Group Convolutions

Equivariant neural networks for SE(3) are often categorized into regular or steerable (tensor field) approaches [Weiler and Cesa, 2019].

- **Regular Group Convolutions:** These typically operate on multi-channel scalar fields defined over a group $G$ or a homogeneous space $X \equiv G/H$ (like $\mathbb{R}^3 \times S^2 \equiv SE(3)/SO(2)$ in our case). Feature fields $f : X \to \mathbb{R}^C$ transform via the regular representation: $[\rho(g)f](x) = f(g^{-1}x)$. Convolutions are then a form of template matching of a kernel $k(g_y^{-1}x)$ with the input signal [Cohen and Welling, 2016, Bekkers, 2019]. A key advantage is that point-wise nonlinearities can be applied directly to these scalar feature maps without breaking equivariance [cf. Bekkers et al., 2024, Appx. A.1].

- **Steerable Group Convolutions / Tensor Field Networks:** These operate on feature fields $f : \mathbb{R}^n \to V_\rho$ where the codomain $V_\rho$ is a vector space carrying a representation $\rho$ of $SO(n)$ (often a sum of irreducible representations, irreps). Features transform via induced representations, affecting both the domain and codomain: $([\mathrm{Ind}_{SO(n)}^{SE(n)} \rho](g)f)(\mathbf{x}) = \rho(\mathbf{R}) f(g^{-1}\mathbf{x})$ [Weiler et al., 2021]. Kernels $k(\mathbf{x})$ must satisfy a steerability constraint $k(\mathbf{Rx}) = \rho_{out}(\mathbf{R})k(\mathbf{x})\rho_{in}(\mathbf{R}^{-1})$. While this allows for exact equivariance without discretizing the rotation group (if using irreps), applying nonlinearities typically requires specialized equivariant operations or transformations to a scalar basis, as standard element-wise activations on steerable features (vectors/tensors) can break equivariance [Weiler and Cesa, 2019].

## B.2   `Rapidash` as a Regular Group Convolution and its Relation to Steerable Networks

`Rapidash`, like PΘNITA [Bekkers et al., 2024], processes feature fields $f : \mathbb{R}^3 \times S^2 \to \mathbb{R}^C$. That is, at each point $\mathbf{x} \in \mathbb{R}^3$, it maintains a scalar signal $f_{\mathbf{x}} : S^2 \to \mathbb{R}^C$ defined over the sphere of orientations $S^2$. This aligns with the regular group convolution paradigm, in which the convolution kernel acts as an $\mathbb{R}^3 \times S^2$, subject to a symmetry constraint due to the quotient space structure $\mathbb{R}^3 \times S^2 \equiv SE(3)/SO(2)$. Specifically, the $SE(3)$ group convolution over $\mathbb{R}^3 \times S^2$ is of the form

$$[Lf](\mathbf{x}, \mathbf{n}) = \iint k(\mathbf{R_n}^T(\mathbf{x}' - \mathbf{x}), \mathbf{R_n}^T \mathbf{n}')f(\mathbf{x}', \mathbf{n}')d\mathbf{x}'d\mathbf{n}', \tag{2}$$

with kernel constraint $\forall R_z \in SO(2) : k(R\mathbf{x}, R\mathbf{n}) = k(\mathbf{x}, \mathbf{n})$ with $R_z$ a rotation around the $z$-axis. This symmetry constraint is automatically solved when conditioning message passing layers (such as convolution layers) on the invariants outlined in [Bekkers et al., 2024, Thm. 1]. In terms of these invariants, the resulting discrete group convolution is given by

$$[Lf](\mathbf{x}_i, \mathbf{n}_i) = \sum_{j \in \mathcal{N}(i)} k(\mathbf{a}_{ij})f(\mathbf{x}_j, \mathbf{n}_j), \tag{3}$$

with the invariant pair-wise attributes given by

$$
\begin{pmatrix} a_{ij}^{(1)} \\ a_{ij}^{(2)} \\ a_{ij}^{(3)} \end{pmatrix} = \begin{pmatrix} \mathbf{n}_i^T(\mathbf{x}_j - \mathbf{x}_i) \\ \|\mathbf{n}_j \perp (\mathbf{x}_j - \mathbf{x}_i)\| \\ \mathbf{n}_i^T \mathbf{n}_i \end{pmatrix} , \tag{4}
$$

with $\perp$ denoting part of the vector $\mathbf{n}_j$ orthogonal to $\mathbf{x}_j - \mathbf{x}_i$.

As detailed in [Bekkers et al., 2024, App A], the connection between regular and steerable convolutions is established through Fourier analysis on the group/homogeneous space [Kondor et al., 2018, Cesa et al., 2021]. A scalar field $f_{\mathbf{x}}(\mathbf{n})$ over $S^2$ (as used in `Rapidash`/`PΘNITA`) can be decomposed into spherical harmonic coefficients (its Fourier transform) through a spherical Fourier transform. These coefficients for different spherical harmonic degrees correspond to features transforming under irreducible representations of SO(3), which are the building blocks of steerable tensor field networks.

**Proposition B.1** (Equivalence via Fourier Transform). *A regular group convolution operating on scalar fields $f : \mathbb{R}^n \times Y \to \mathbb{R}^C$ (where $Y$ is $S^{n-1}$ or $SO(n)$) can be equivalently implemented as a steerable group convolution operating on fields of Fourier coefficients $\hat{f} : \mathbb{R}^n \to V_\rho$ (where $V_\rho$ is the space of combined irreducible representations) by performing point-wise Fourier transforms $\mathcal{F}_Y$ before the steerable convolution and inverse Fourier transforms $\mathcal{F}_Y^{-1}$ after.*

**Remark 2.** *This equivalence is discussed in depth by Bekkers et al. [2024, Appx. A.1], Brandstetter et al. [2022], and Cesa et al. [2021]. Consequently, `Rapidash` could, in principle, be reformulated as a tensor field network by operating in the spherical harmonic (Fourier) domain.*

## B.3 Universal Approximation

The universal approximation capabilities of equivariant networks are crucial. For steerable tensor field networks, Dym and Maron [2020] proved universal approximation properties for equivariant graph neural networks. Building on such results, and the correspondence between regular and steerable views, it has been shown that message passing networks (which include architectures like `Rapidash`) conditioned on appropriate invariant attributes over position-orientation space ($\mathbb{R}^n \times S^{n-1}$) are equivariant universal approximators.

**Corollary B.1** (Universal Approximation for Rapidash). *`Rapidash`, as an instance of message passing networks over $\mathbb{R}^3 \times S^2$ with message functions conditioned on the bijective invariant attributes (derived in Bekkers et al. [2024, Thm. 1]), is an SE(3)-equivariant universal approximator.*

**Remark 3.** *This follows from Bekkers et al. [2024, Cor. 1.1], which itself builds on universality results for steerable GNNs [Dym and Maron, 2020] and for invariant networks used to construct equivariant functions [Villar et al., 2021, Gasteiger et al., 2021]. The key is that feature maps over $\mathbb{R}^3 \times S^2$ are sufficiently expressive.*

## B.4 Advantages of the Regular Group Convolution Viewpoint for Rapidash

`Rapidash` adopts the regular group convolution viewpoint, working with scalar signals on discretized spherical fibers ($f(\mathbf{x}, \mathbf{n}_k)$ where $\mathbf{n}_k$ are grid points on $S^2$). This offers practical advantages:

1. **Simplicity of Activation Functions:** Since the features $f(\mathbf{x}, \mathbf{n}_k)$ at each grid point are scalars (or vectors of scalars in the channel dimension), standard element-wise nonlinear activation functions (e.g., GELU, ReLU, SiLU) can be applied directly without breaking SE(3)-equivariance. This is because the action of $g \in$ SE(3) permutes these values on the fiber or spatially, but the activation acts on each scalar value independently. In contrast, steerable tensor field networks require specialized equivariant nonlinearities or norm-based activations on higher-order tensors to preserve equivariance [Weiler and Cesa, 2019], which can limit expressivity or introduce computational overhead.

2. **Computational Efficiency of Activations:** While steerable networks *can* apply scalar activations by first performing an inverse Fourier transform (to get scalar fields), applying the activation, and then a forward Fourier transform (back to irreps), this incurs significant computational cost at each nonlinearity [cf. Bekkers et al., 2024, Appx. A.1]. By operating directly on scalar spherical signals, `Rapidash` avoids these repeated transformations. Previous work on steerable group convolutions has indeed found that element-wise activation

functions applied to scalar fields (obtained via inverse Fourier transforms from steerable vector features) can be most effective [e.g., as implicitly done in some equivariant GNNs by taking norms or scalar products before activation, or as explicitly discussed for general steerable CNNs in Weiler and Cesa, 2019].

3. **Conceptual Simplicity:** The regular group convolution approach, involving template matching of kernels over signals on $G/H$, can be more intuitive and closer to standard CNN paradigms than navigating representation theory and Clebsch-Gordan tensor products often required for constructing steerable tensor field networks. Concepts like stride/sub-sampling and normalization layers readily transfer to this setting.

While discretizing the sphere $S^2$ introduces an approximation to full SO(3) equivariance (equivariance up to the grid resolution), empirical results, including those for PΘNITA [Bekkers et al., 2024] and our findings with `Rapidash`, demonstrate that this is not detrimental to achieving state-of-the-art performance and robust generalization.

### B.5 Separable group convolutions

Regular group convolutions over the full spacew $SE(3)$ can be efficiently computed when the kernel is factorized via

$$k_{c'c}(\mathbf{x}, \mathbf{R}) = k_c^{\mathbb{R}^3}(\mathbf{x}) k_c^{\text{SO}(3)}(\mathbf{R}) k_{c'c}^{(channel)},$$

with $c, c'$ the row and column indices of the "channel mixing" matrix. Then the group convolution equation can be split into three steps that are each efficient to compute: a spatial interaction layer (message passing), a point-wise SO(3) convolution, and a point-wise linear layer [Knigge et al., 2022, Kuipers and Bekkers, 2023]. It would result in the group convolutional counterpart of *depth-wise separable convolution* [Chollet, 2017], which separates convolution in two steps (spatial mixing and channel mixing). In particular, for our choice to the the group convolution over $\mathbb{R}^3 \times S^2$ using the pair-wise invariants of (4) the kernel is parametrized as

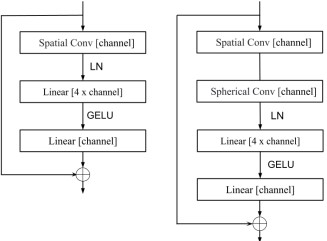

Figure 6: Block design with base space $\mathbb{R}^3 \times S^2$.

$$k_{c'c}(\mathbf{a}_{ij}) = k_c^{\mathbb{R}^3}(a_{ij}^{(1)}, a_{ij}^{(2)}) k_c^{S^2}(a_{ij}^{(3)}) k_{c'c}^{(channel)} .$$

This form allows to split the convolution over several steps, which following [Bekkers et al., 2024] we adapt a ConvNext [Liu et al., 2022] as a the main layer to parametrize `Rapidash`. Fig. 6 shows the the steps performed in this block, relative a standard ConvNext block over position space only. Here LN denote layer norm and GELU is used as activation function.

## C Sources of Equivariance Breaking

In the study of equivariant deep learning, it is important to distinguish between different notions of "breaking" symmetries. Some approaches, such as those explored by Lawrence et al. [2025a] or the pose-conditioning methods analyzed in Appendix D, utilize architectures that maintain specific (joint) equivariance properties to achieve "symmetry breaking" in the output (e.g., an output sample being less symmetric than the input, or overcoming the limitations of standard invariance). This section, in contrast, focuses on mechanisms by which the strict $G$-equivariance of a neural network architecture itself can be compromised or intentionally relaxed, leading to a model that no longer fully adheres to the mathematical definition of $G$-equivariance. We categorize these into *external* and *internal* sources of equivariance breaking.

### C.1 External Equivariance Breaking

External equivariance breaking occurs when an inherently $G$-equivariant architecture loses its equivariance properties due to the way inputs are provided to the network. Consider a linear layer $L$ designed to be $G$-equivariant (e.g., for $G = \text{SE}(3)$). Let $v$ be a vector input that transforms naturally under the group action, and define $x_g = g \cdot v$ as its transformed version, described in global coordinates. When these coordinates $x_g$ are provided, for example, as a set of scalar triplets rather than

as a geometric vector type that the layer is designed to process equivariantly, they may be treated independently by the network without regard to their collective transformation under $G$. In such cases, we have:

$$\text{if } x \neq x_g, \text{ then often } L(x_g) \neq g \cdot L(x) \text{ (or } L(x_g) \neq L(x) \text{ for invariant L)} \tag{5}$$

This means the network processes transformed inputs in a way that does not respect the group symmetry, thereby breaking the intended $G$-equivariance of the function $L$. In contrast, when inputs are specified as types that transform appropriately under the group action (e.g., as vectors for an $SE(3)$-equivariant layer that expects vector inputs), equivariance can be maintained:

$$L(g \cdot v) = g \cdot L(v) \quad \forall g \in G \tag{6}$$

Moreover, for features that are truly invariant under $G$ (such as one-hot encodings of atom types in QM9, which are invariant to $SO(3)$ rotations), the group action is trivial:

$$g \cdot x_{inv} = x_{inv} \quad \implies \quad L(g \cdot x_{inv}) = L(x_{inv}) \tag{7}$$

This ensures that processing these features does not break the network's overall desired invariance to group transformations of other, non-invariant inputs.

## C.2 Internal Equivariance Breaking

Internal equivariance breaking refers to the deliberate relaxation or incorrect specification of equivariance constraints within the layers themselves. Even if inputs are provided correctly, the layer operations may not fully respect the symmetry group $G$. Recent works have explored various approaches to controlled relaxation, including [Wang et al., 2022]:

- Basis decomposition methods that mix $G$-equivariant and non-$G$-equivariant components within a layer.
- Learnable deviations from strict $G$-equivariance, often controlled by regularization terms.
- Progressive relaxation or enforcement of $G$-equivariance constraints during the training process.

In some cases, an internally broken layer $L_{\text{broken}}$ might be expressed as a combination of a strictly $G$-equivariant part and a non-$G$-equivariant part:

$$L_{\text{broken}}(X) = L_{\text{equiv}}(X) + \alpha L_{\text{non-equiv}}(X) \tag{8}$$

where $\alpha$ controls the degree of deviation from strict $G$-equivariance. Some approaches (e.g., [Wang et al., 2022]) implement schemes where $\alpha$ is annealed during training.

## C.3 Interplay of Equivariance Breaking Mechanisms

In practice, both external and internal equivariance-breaking can occur, sometimes simultaneously, and their effects can interact. For example, a network might employ layers with relaxed internal constraints (e.g., non-stationary filters) while also processing geometric inputs (like coordinates) in a manner that externally breaks the intended global symmetry (e.g., treating them as independent scalar features). The overall adherence of the model to a specific group symmetry $G$ then depends on the interplay of these factors. As noted by Petrache and Trivedi [2023], while aligning data symmetry with architectural symmetry (strict equivariance) is ideal, models with partial or approximate equivariance can still exhibit improved generalization compared to fully non-equivariant ones.

## C.4 Relevance to Architectural Choices in Our Study

While our study primarily investigates scenarios of *external equivariance breaking* (e.g., how providing coordinate information as scalars versus vectors impacts overall $SE(3)$ equivariance, cf. the red exclamation marks in our tables), some of our architectural comparisons also touch upon concepts related to the scope of equivariance. For instance, choosing to build a model with strictly $T_3$-equivariant layers (translation-equivariant only, Tab. 2, rows 15-18) instead of $SE(3)$-equivariant layers (Table 1, rows 1-4) results in a model that is not $SE(3)$-equivariant overall. This is an architectural design choice selecting a different, less encompassing symmetry group, rather than starting with an $SE(3)$-layer and internally relaxing its $SE(3)$ constraints.

Our experiments demonstrate that carefully managing how equivariance is implemented or broken, whether externally through input representation or by choosing specific architectural symmetry properties, can significantly impact model performance, especially when the underlying data exhibits only approximate symmetries or when task-relevant information is tied to a canonical orientation. This aligns with recent findings showing benefits from various approaches to relaxed or modified equivariance constraints [van der Ouderaa et al., 2022, Kim et al., 2023, Pertigkiozoglou et al., 2024]. Further theoretical analysis of how explicitly providing pose information can be seen as a principled modification of standard invariance is presented in Appendix D.

### C.5 Relationship to Our Work

While our study primarily focuses on external symmetry breaking (cf. the red exclamation marks in our tables), we note that the transition between translation-equivariant layers (Table 2, rows 15-18) and roto-translation equivariant layers (Table 1, rows 1-4) could be viewed as a form of internal symmetry breaking. However, we distinguish our approach from explicit internal symmetry-breaking methods as we do not implement continuous relaxation of equivariance constraints within layers, but rather compare discrete architectural choices with different symmetry properties.

## D  Analysis of Symmetry Breaking and Pose-Conditioning in Equivariant Models

### D.1  Standard Invariance and Effective Pose Entropy

This section formalizes the notion that a standard $SO(3)$-invariant model, when viewed within the framework of a jointly invariant function $f(X, Z)$, operates as if the auxiliary pose variable $Z$ carries no specific information, i.e., it corresponds to a maximum entropy distribution over poses.

**Proposition D.1** (Standard $SO(3)$-Invariance as Maximum Effective Pose Entropy). *Let $f(X, Z)$ be a function $f : \mathcal{X} \times SO(3) \to Y$ that is jointly $SO(3)$-invariant, meaning $f(gX, gZ) = f(X, Z)$ for all $g \in SO(3)$. If the output of $f(X, Z)$ is also required to be standard $SO(3)$-invariant with respect to $X$ alone, defining a function $f_{inv}(X)$ such that $f_{inv}(gX) = f_{inv}(X)$ for all $g \in SO(3)$, then:*

1. *The function $f(X, Z)$ must be independent of the auxiliary pose variable $Z$. Specifically, $f(X, Z) = f(X, Id)$ for any $Z \in SO(3)$, where $Id$ is the identity element in $SO(3)$.*

2. *Consequently, such a model $f(X, Z)$ (which produces $f_{inv}(X)$) behaves as if $Z$ is drawn from an uninformative, maximum entropy distribution (e.g., the uniform distribution over $SO(3)$). The model cannot utilize any specific orientation information conveyed by $Z$.*

*Proof Sketch.* To demonstrate part 1, that $f(X, Z) = f(X, Id)$:

1. By the joint $SO(3)$-invariance of $f$, we have $f(X, Z) = f(Z^{-1}X, Z^{-1}Z) = f(Z^{-1}X, Id)$.

2. The condition that the output of $f(X, Z)$ is standard $SO(3)$-invariant with respect to $X$ means that for any fixed second argument (like $Id$), the function $f(\cdot, Id)$ must be $SO(3)$-invariant in its first argument. That is, $f(gX, Id) = f(X, Id)$ for all $g \in SO(3)$.

3. Applying this standard $SO(3)$-invariance with $g = Z^{-1}$ to the expression $f(Z^{-1}X, Id)$, we get $f(Z^{-1}X, Id) = f(X, Id)$.

4. Combining steps 1 and 3: $f(X, Z) = f(Z^{-1}X, Id) = f(X, Id)$.

This establishes that $f(X, Z)$ is independent of $Z$.

For part 1, if $f(X, Z)$ is independent of $Z$, it cannot make use of any particular value of $Z$ to alter its output. From an informational perspective, $Z$ provides no specific information to the model. This behavior is equivalent to $Z$ being drawn from a distribution that reflects maximal uncertainty about the pose, which for a compact group like $SO(3)$ is the uniform (Haar) measure, corresponding to maximum entropy. Thus, the model $f_{inv}(X)$ cannot utilize any specific orientation information that might be notionally carried by $Z$. $\square$

## D.2 Proof of Corollary 3.1 (Expressivity Gain from Low-Entropy Pose Information)

**Corollary** 3.1 states: *Let the optimal invariant mapping for a task, $f^*(X, R)$, depend non-trivially on canonical orientation $R$. Based on the distinct informational content of $Z$ outlined above: (a) Standard invariant models $f_{inv}(X)$ (maximum effective pose entropy) lack the expressivity to represent $f^*$; and (b) Pose-conditioned models $f_{cond}(X, R)$ (provided with zero-entropy pose information) can represent $f^*$.*

Let $G = SO(3)$ be the group of orientations. The optimal mapping $f^* : \mathcal{X} \times G \to \mathcal{Y}$ (where $\mathcal{X}$ is the space for $X$ and $\mathcal{Y}$ for the output) has two key properties:

1. **Joint Invariance for an Invariant Task:** $f^*(gX, gR) = f^*(X, R)$ for all $g \in G$.

2. **Non-trivial Dependence on $R$:** For any given $X \in \mathcal{X}$, there exist $R_1, R_2 \in G$ such that $R_1 \neq R_2$ but $f^*(X, R_1) \neq f^*(X, R_2)$. This implies that $R$ is an essential input for determining the output of $f^*$, not merely a redundant pose of $X$ that could be factored out by invariance.

**Proof of 1: Standard invariant models $f_{inv}(X)$ lack the expressivity to represent $f^*$.**

A standard $G$-invariant model $f_{inv}(X)$ is a function $f_{inv} : \mathcal{X} \to \mathcal{Y}$. By definition, its output depends solely on $X$ and must satisfy $f_{inv}(gX) = f_{inv}(X)$ for all $g \in G$. For any specific input $X_0 \in \mathcal{X}$, $f_{inv}(X_0)$ produces a single, uniquely determined output value, say $Y_{inv\_0}$.

The model $f_{inv}(X)$ does not take $R$ as an explicit input. As established by the principle that standard invariant models operate with maximum effective entropy regarding an auxiliary pose variable (see Appendix D.1, Proposition D.1), $f_{inv}(X)$ cannot vary its output based on different values of $R$ for a fixed $X$. Its output is fixed once $X$ is fixed.

Now, consider the target function $f^*(X_0, R)$. According to property (2) of $f^*$, there exist distinct $R_1, R_2 \in G$ such that $f^*(X_0, R_1) = Y_1^*$ and $f^*(X_0, R_2) = Y_2^*$, where $Y_1^* \neq Y_2^*$.

If $f_{inv}(X)$ were to represent $f^*(X, R)$, then for the input $X_0$, it would need to output $Y_1^*$ when the canonical orientation is $R_1$, and simultaneously output $Y_2^*$ when the canonical orientation is $R_2$. However, $f_{inv}(X_0)$ can only produce its single output $Y_{inv\_0}$. Since $Y_1^* \neq Y_2^*$, $Y_{inv\_0}$ cannot equal both. Therefore, $f_{inv}(X)$ cannot represent the function $f^*(X, R)$ due to its inability to differentiate its output based on $R$.

**Proof of 2: Pose-conditioned models $f_{cond}(X, R)$ *can* represent $f^*$.**

A pose-conditioned model $f_{cond}(X, R)$ is a function $f_{cond} : \mathcal{X} \times G \to \mathcal{Y}$. It explicitly takes both $X$ and $R$ as inputs. It is designed to satisfy the same joint $G$-invariance as the target: $f_{cond}(gX, gR) = f_{cond}(X, R)$.

The target function $f^*(X, R)$ is a specific function mapping from the domain $\mathcal{X} \times G$ to $\mathcal{Y}$ that adheres to this joint $G$-invariance. We assume that the class of models from which $f_{cond}(X, R)$ is drawn (e.g., sufficiently large neural networks architected to respect joint $G$-invariance) are universal approximators for continuous functions on $\mathcal{X} \times G$ that satisfy the given symmetry requirements.

Since $f_{cond}(X, R)$ takes $R$ as an explicit input (representing zero-entropy pose information for that instance, as $R$ is known and fixed), it has the architectural capacity to make its output depend on $R$. For a given $X_0$, the model $f_{cond}(X_0, R)$ can learn to produce different outputs for different values of $R$. Specifically, it can learn to output $f^*(X_0, R_1)$ when its input $R$ is $R_1$, and $f^*(X_0, R_2)$ when its input $R$ is $R_2$.

Given that $f^*(X, R)$ is a well-defined function of $(X, R)$ satisfying the joint $G$-invariance, and $f_{cond}(X, R)$ is a universal approximator for such functions with access to both $X$ and $R$, $f_{cond}(X, R)$ can represent $f^*(X, R)$.

## D.3 On Symmetry Breaking in ShapeNet Segmentation

Providing explicit canonical pose information $R$ to our segmentation models acts as a crucial form of symmetry breaking. This allows models to overcome limitations inherent in standard $SO(3)$-equivariant networks, $f_{std\_eq}(X)$, which operate solely on an input point cloud $X$. Such standard models are bound by Curie's Principle, meaning their output segmentation must respect

any symmetries present in $X$. Consequently, if $X$ possesses exact internal symmetries (e.g., a $C_4$-symmetric table) or even strong approximate or coarse-level symmetries (e.g., an airplane viewed broadly as a cross-shape), $f_{std\_eq}(X)$ can struggle to distinguish parts that are (nearly) equivalent under these symmetries but are semantically distinct with respect to a canonical frame (like specific table legs or differentiating "top" from "bottom" on an airplane). These (approximate) symmetries can lead to ambiguities that $f_{std\_eq}(X)$ cannot resolve using only the information in $X$.

By contrast, a pose-conditioned model, $f_{cond\_eq}(X, R)$, receives $X$ alongside its known canonical pose reference $R$ (e.g., $R = I$ for aligned ShapeNet data). The model's joint $SO(3)$-equivariance now applies to the effective input pair $(X, R)$. Critically, the specific orientation $R$ typically makes this pair $(X, R)$ maximally asymmetric with respect to $SO(3)$ transformations (i.e., its symmetry group $G_{(X,R)}$ becomes trivial, $\{I\}$), even if $X$ itself has notable exact or approximate symmetries. Because the effective input is asymmetric, $f_{cond\_eq}(X, R)$ is no longer constrained by $X$'s original symmetries and can produce segmentations that make distinctions based on $R$. For instance, it can learn to identify the "front-left leg" of a symmetric table or the "upper surface" of an airplane wing, as $R$ provides the necessary external frame to dissolve these ambiguities.

This ability to generate more specific, $R$-conditioned segmentations by resolving ambiguities arising from $X$'s exact or approximate symmetries represents a significant expressivity gain, as supported by the logic of 'Corollary 3.1' (adapted for equivariant outputs). The model $f_{cond\_eq}(X, R)$ can represent optimal segmentation functions $f_{seg}^*(X, R)$ that depend on the canonical frame $R$, which $f_{std\_eq}(X)$ cannot. This deterministic disambiguation via $R$ allows the model to overcome symmetry-induced limitations, distinct from, yet achieving a similar practical outcome to, the probabilistic symmetry breaking described by Lawrence et al. [2025b].

## D.4 Symmetry Breaking in Generative Modeling of ShapeNet Objects

Generative modeling for ShapeNet aims to map a symmetric base distribution $p_0(\mathbf{z})$ (e.g., $SO(3)$-invariant Gaussian noise) to the target distribution $p_1(\mathbf{x})$ of canonically aligned ShapeNet objects. This target $p_1(\mathbf{x})$ is not $SO(3)$-invariant. Standard $SO(3)$-equivariant generative processes would inherently preserve the $SO(3)$-symmetry of $p_0(\mathbf{z})$ throughout the generation. By Curie's Principle, an equivariant map cannot produce a less symmetric output distribution from a more symmetric input distribution. Thus, a purely $SO(3)$-equivariant denoising function $D_\theta(\mathbf{x}_t, t)$ (or associated score, or flow [Karras et al., 2022]) would generate an $SO(3)$-invariant distribution of shapes, failing to match the aligned ShapeNet data.

To resolve this, symmetry must be broken. This is achieved by conditioning the denoising model on explicit pose information $R$, yielding $D_\theta(\mathbf{x}_t, t, R)$. For generating aligned ShapeNet objects, $R$ is fixed to a canonical pose (e.g., $R = I$). The model $D_\theta(\mathbf{x}_t, t, R)$ is designed for joint $SO(3)$-equivariance (i.e., $D_\theta(g\mathbf{x}_t, t, gR) = gD_\theta(\mathbf{x}_t, t, R)$).

By providing a fixed $R_{target}$ (e.g., $R = I$), the effective input to the jointly equivariant model becomes $(\mathbf{x}_t, R_{target})$. This pair can be considered maximally asymmetric (its symmetry group $G_{(\mathbf{x}_t, R_{target})}$ is often trivial, as discussed in Appendix D.3). Consequently, the model can map potentially symmetric noise $\mathbf{x}_t$ towards samples $\mathbf{x}_0$ specifically aligned with $R_{target}$. The fixed pose $R_{target}$ breaks the initial $SO(3)$-symmetry, enabling the generation of the non-$SO(3)$-invariant target distribution. This use of pose $R$ aligns with the principle from Lawrence et al. [2025b] that auxiliary information can enable equivariant networks to produce outputs less symmetric than their primary inputs.

## D.5 Generalization Benefits of Joint Invariance in Pose-Conditioned Models

While App. D established the expressivity advantage of pose-conditioned invariant models $f_{cond}(X, R)$ over standard invariant models $f_{inv}(X)$, one might consider if a sufficiently complex non-equivariant model $f_{neq}(X, R)$, which also takes pose $R$ as input but lacks structural symmetry constraints, could achieve similar performance. Assuming both $f_{cond}(X, R)$ (satisfying joint invariance $f(gX, gR) = f(X, R)$) and $f_{neq}(X, R)$ have sufficient capacity to represent the optimal invariant solution $f^*(X, R)$, we argue that the equivariant structure of $f_{cond}$ provides superior generalization guarantees.

This argument leverages theoretical results on the generalization benefits derived from incorporating known symmetries, such as Theorem 6.1 presented by Lawrence et al. [2025a], which extends foundational work on equivariance and generalization [Elesedy and Zaidi, 2021].

**Proposition D.2** (Generalization Advantage of Jointly Invariant Pose-Conditioned Models). *Let $X$ be drawn from a $G$-invariant distribution $\mathbb{P}(X)$, where $G = SO(3)$. Let $R$ be the provided pose, treated as an auxiliary variable $Z = R$ with the equivariant conditional distribution $\mathbb{P}(Z|X) = \delta(Z - R)$. Consider two models predicting an invariant output $Y$ using inputs $(X, R)$:*

- *$f_{cond}(X, R)$: A model satisfying joint invariance, $f_{cond}(gX, gR) = f_{cond}(X, R)$.*

- *$f_{neq}(X, R)$: Any model using the same inputs $(X, R)$ that does **not** satisfy joint invariance.*

*Under suitable regularity conditions (as specified in Lawrence et al. [2025a], Thm 6.1) and for a suitable risk function $R(f) = \mathbb{E}[\mathcal{L}(f(X, R), Y)]$ (e.g., using $L_2$ loss), the expected risk of the non-equivariant model is greater than or equal to that of the jointly invariant model:*

$$R(f_{neq}) \geq R(f_{cond})$$

*The difference $R(f_{neq}) - R(f_{cond})$ represents a non-negative generalization gap attributable to the component of $f_{neq}$ orthogonal to the space of jointly invariant functions.*

**Remark 4.** *This proposition follows from applying Theorem 6.1 [Lawrence et al., 2025a] to our setting. The conditions are met: $\mathbb{P}(X)$ is $G$-invariant (typically assumed or achieved via augmentation), $\mathbb{P}(Z|X) = \delta(Z - R)$ is equivariant, and $G$ acts freely on $Z = R \in SO(3)$.*

Proposition D.2 provides formal support for preferring the pose-conditioned equivariant (jointly invariant) architecture $f_{cond}(X, R)$ over an unstructured, non-equivariant model $f_{neq}(X, R)$ that uses the same input information. By incorporating the known relationship between transformations of the input $X$ and the pose $R$ via the joint invariance constraint, the equivariant model leverages a useful inductive bias. This bias restricts the hypothesis space to functions consistent with the underlying geometry, thereby reducing variance and improving expected generalization performance compared to a less constrained model, even if both models possess sufficient capacity to represent the optimal solution.

# E  Additional Results and Discussion

## E.1  General experiments

**Results with extra channels** In this section, we present additional results to support the hypothesis presented in the paper. We expand ShapeNet3D table in the main paper with additional models from literature in Tab.4 and Tab.5, we show an ablation study on the ShapeNet3D dataset for the part segmentation task as well as the CMU motion capture dataset for the human motion prediction task. Here we present results with two different settings of hidden features, C = 256 (gray) and C = 2048. The latter inflated model was trained to match the representation capacity of the rest of the models.

**Classification task on Modelnet40 dataset** ModelNet40 dataset [Wu et al., 2015] contains 9,843 training and 2,468 testing meshed CAD models belonging to 40 categories. We present Modelnet40 classification results in Tab. 6.

**Practical implications of the results** To get insights for building models on real data, we perform experiments on ModelNet40 on the classification task. Our experiments demonstrate similar trends to those discussed in the main paper for the experiments on ShapeNet3D, QM9, and CMU motion datasets. For a non-equivariant task such as classification, we see that equivariant methods perform best (model 8 in Tab. 6), models with equivariance breaking (model 4 and 15) as well as non-equivariant models (model 19) perform almost equally well, and the performance gap is small.

**Scaling experiments** To show the effect of adding more training data, we perform ShapeNet3D segmentation for $60\%, 80\%, 100\%$ of the data. The gap in performance is more for models with effective equivariance $T_3$ as compared to effective equivariance $SO(3)$ and $SE(3)$. See Tab. 7.

## E.2  Fixed network capacity comparisons on QM9 generation and regression

This section provides a detailed investigation into the interplay of model capacity, computational efficiency, and the efficacy of different group-equivariant inductive biases on the QM9 molecular

Table 4: Comparing models with inflated representation capacity against regular models on ShapeNet3D for part segmentation task with two different settings of hidden features, C = 256 (gray) and C = 2048.

| | | Coordinates as Scalars | Coordinates as Vectors | Normals as Scalars | Normals as Vectors | Global Frame | Effective Equivariance | IoU↑ (aligned) | IoU↑ (rotated) | Normalized Epoch Time |
|---|---|---|---|---|---|---|---|---|---|---|
| Rapidash with internal SE(3) Equivariance Constraint | | | | | | | | | | |
| Model Variation | Type | | | | | | | | | |
| 1 | $\mathbb{R}^3$ | ✗ | - | ✗ | - | - | SE(3) | $80.26_{\pm0.06}$ $80.31_{\pm0.15}$ | $\mathbf{80.33}_{\pm0.13}$ $80.20_{\pm0.07}$ | $\sim 1$ $\sim 10$ |
| 2 | $\mathbb{R}^3$ | ✗ | - | ✓ | - | - | $T_3$ | $83.95_{\pm0.06}$ $83.87_{\pm0.09}$ | $52.09_{\pm0.87}$ $49.63_{\pm0.87}$ | $\sim 1$ $\sim 10$ |
| 3 | $\mathbb{R}^3$ | ✓ | - | ✗ | - | - | none | $84.23_{\pm0.08}$ $84.01_{\pm0.06}$ | $34.15_{\pm0.05}$ $32.22_{\pm0.27}$ | $\sim 1$ $\sim 10$ |
| 4 | $\mathbb{R}^3$ | ✓ | - | ✓ | - | - | none | $\mathbf{84.75}_{\pm0.02}$ $84.48_{\pm0.16}$ | $34.07_{\pm0.43}$ $32.90_{\pm0.47}$ | $\sim 1$ $\sim 10$ |
| Rapidash with Internal $T_3$ Equivariance Constraint | | | | | | | | | | |
| 16 | $\mathbb{R}^3$ | ✗ | - | ✗ | - | - | $T_3$ | $85.26_{\pm0.01}$ $85.38_{\pm0.05}$ | $31.41_{\pm0.86}$ $32.78_{\pm0.59}$ | $\sim 1$ $\sim 10$ |
| 17 | $\mathbb{R}^3$ | ✗ | - | ✓ | - | - | $T_3$ | $\mathbf{85.82}_{\pm0.10}$ $85.71_{\pm0.17}$ | $\mathbf{35.42}_{\pm0.98}$ $32.25_{\pm0.07}$ | $\sim 1$ $\sim 10$ |
| 18 | $\mathbb{R}^3$ | ✓ | - | ✗ | - | - | none | $85.51_{\pm0.06}$ $85.52_{\pm0.09}$ | $32.79_{\pm0.74}$ $31.11_{\pm0.15}$ | $\sim 1$ $\sim 10$ |
| 19 | $\mathbb{R}^3$ | ✓ | - | ✓ | - | - | none | $85.66_{\pm0.02}$ $85.16_{\pm0.06}$ | $33.55_{\pm0.49}$ $30.62_{\pm0.21}$ | $\sim 1$ $\sim 10$ |
| Reference methods from literature | | | | | | | | | | |
| DeltaConv | - | - | - | - | - | - | - | 86.90 | - | - |
| PointNeXt | - | - | - | - | - | - | - | 87.00 | - | - |
| GeomGCNN * | - | - | - | - | - | - | - | 89.10 | - | - |

Table 5: Ablation study on Human motion prediction task on CMU Motion Capture dataset with inflated models and regular models with two different settings of hidden features, C = 256 (gray) and C = 2048. We evaluate our models using the mean squared error ($\times 10^{-2}$) metric.

| | | Coordinates as Scalars | Coordinates as Vectors | Velocity as Scalars | Velocity as Vectors | Global Frame | Effective Equivariance | MSE | MSE (rotated) | Normalized Epoch Time |
|---|---|---|---|---|---|---|---|---|---|---|
| Rapidash with internal $SE(3)$ Equivariance Constraint | | | | | | | | | | |
| Model Variation | Type | | | | | | | | | |
| 1 | $\mathbb{R}^3$ | ✗ | - | ✗ | - | - | SE(3) | $> 100$ $> 100$ | $> 100$ $> 100$ | $\sim 1$ $\sim 20$ |
| 2 | $\mathbb{R}^3$ | ✗ | - | ✓ | - | - | $T_3$ | $5.44_{\pm0.12}$ $\mathbf{4.81}_{\pm0.13}$ | $24.45_{\pm0.66}$ $25.66_{\pm1.53}$ | $\sim 1$ $\sim 20$ |
| 3 | $\mathbb{R}^3$ | ✓ | - | ✗ | - | - | none | $6.88$ $5.93$ | $> 100$ $84.54$ | $\sim 1$ $\sim 20$ |
| 4 | $\mathbb{R}^3$ | ✓ | - | ✓ | - | - | none | $5.93$ $5.53$ | $> 100$ $> 100$ | $\sim 1$ $\sim 19$ |
| Rapidash with Internal $T_3$ Equivariance Constraint | | | | | | | | | | |
| 15 | $\mathbb{R}^3$ | ✗ | - | ✗ | - | - | $T_3$ | $6.53$ $5.99$ | $43.80$ $> 100$ | $\sim 1$ $\sim 19$ |
| 16 | $\mathbb{R}^3$ | ✗ | - | ✓ | - | - | $T_3$ | $\mathbf{5.3}_{\pm0.04}$ $24.32_{\pm1.97}$ | $\mathbf{40.69}_{\pm0.87}$ $> 100_{\pm6.65}$ | $\sim 1$ $\sim 19$ |
| 17 | $\mathbb{R}^3$ | ✓ | - | ✗ | - | - | none | $6.03$ $6.82$ | $77.78$ $69.42$ | $\sim 1$ $\sim 19$ |
| 18 | $\mathbb{R}^3$ | ✓ | - | ✓ | - | - | none | $5.49$ $5.54$ | $66.47$ $76.12$ | $\sim 1$ $\sim 19$ |
| Reference methods from literature | | | | | | | | | | |
| TFN | - | - | - | - | - | - | - | 66.90 | - | - |
| EGNN | - | - | - | - | - | - | - | 31.70 | - | - |
| CGENN | - | - | - | - | - | - | - | 9.41 | - | - |
| CSMPN | - | - | - | - | - | - | - | 7.55 | - | - |

tasks, complementing the findings in Section 5. Specifically, we compare three model variants that represent a spectrum of constraints on the $\mathbb{R}^3 \times \mathbb{S}^2$ base space, keeping their total representational capacity ($O \times C$) and therefore training/inference time approximately constant:

1. **None ($\mathbb{R}^3$ Fully Unconstrained)**: Model with no explicit symmetry constraint on the kernel, $k(\mathbf{x}_i, \mathbf{x}_j)$. The kernel depends on both the receiving point $\mathbf{x}_i$ and the sending point $\mathbf{x}_j$, taking a 6-dimensional input, representing the most unconstrained hypothesis space.

2. **$T_3$ (Translation Equivariant)**: Model with a translation-equivariant constraint, $k(\mathbf{x}_i, \mathbf{x}_j) = k(\mathbf{x}_j - \mathbf{x}_i)$. The kernel depends only on the relative position, taking a 3-dimensional input.

Table 6: Ablation study on ModelNet40 for classification task using accuracy as metric.

| | | Rapidash with internal SE(3) Equivariance Constraint | | | | | | |
|---|---|---|---|---|---|---|---|---|
| Model Variation | Type | Coordinates as Scalars ! | Coordinates as Vectors ! | Normals as Scalars ! | Normals as Vectors | Global Frame ! | Effective Equivariance | Accuracy (%) |
| 1 | $\mathbb{R}^3$ | ✗ | - | ✗ | - | - | SE(3) | $71.04_{\pm 0.79}$ |
| 2 | $\mathbb{R}^3$ | ✗ | - | ✓ | - | - | $T_3$ | $86.75_{\pm 0.12}$ |
| 3 | $\mathbb{R}^3$ | ✓ | - | ✗ | - | - | none | $86.20_{\pm 0.29}$ |
| 4 | $\mathbb{R}^3$ | ✓ | - | ✓ | - | - | none | $\mathbf{88.68}_{\pm 0.24}$ |
| 5 | $\mathbb{R}^3 \times S^2$ | ✗ | ✗ | ✗ | ✗ | ✗ | SE(3) | $85.14_{\pm 0.28}$ |
| 6 | $\mathbb{R}^3 \times S^2$ | ✗ | ✗ | ✗ | ✗ | ✓ | SE(3) | $88.52_{\pm 0.49}$ |
| 7 | $\mathbb{R}^3 \times S^2$ | ✗ | ✗ | ✗ | ✓ | ✗ | SE(3) | $86.06_{\pm 0.57}$ |
| 8 | $\mathbb{R}^3 \times S^2$ | ✗ | ✗ | ✗ | ✓ | ✓ | SE(3) | $\mathbf{89.61}_{\pm 0.25}$ |
| 9 | $\mathbb{R}^3 \times S^2$ | ✗ | ✓ | ✗ | ✗ | ✗ | SO(3) | $85.94_{\pm 0.11}$ |
| 10 | $\mathbb{R}^3 \times S^2$ | ✗ | ✓ | ✗ | ✗ | ✓ | SO(3) | $89.09_{\pm 0.31}$ |
| 11 | $\mathbb{R}^3 \times S^2$ | ✗ | ✓ | ✗ | ✓ | ✗ | SO(3) | $85.41_{\pm 2.17}$ |
| 12 | $\mathbb{R}^3 \times S^2$ | ✗ | ✓ | ✗ | ✓ | ✓ | SO(3) | $89.13_{\pm 0.13}$ |
| 13 | $\mathbb{R}^3 \times S^2$ | ✗ | ✗ | ✓ | ✗ | ✗ | $T_3$ | $86.98_{\pm 0.69}$ |
| 14 | $\mathbb{R}^3 \times S^2$ | ✓ | ✗ | ✗ | ✗ | ✗ | none | $87.25_{\pm 1.07}$ |
| 15 | $\mathbb{R}^3 \times S^2$ | ✓ | ✗ | ✓ | ✗ | ✗ | none | $\mathbf{88.88}_{\pm 0.18}$ |
| | | Rapidash with Internal $T_3$ Equivariance Constraint | | | | | | |
| 16 | $\mathbb{R}^3$ | ✗ | - | ✗ | - | - | $T_3$ | $87.55_{\pm 0.12}$ |
| 17 | $\mathbb{R}^3$ | ✗ | - | ✓ | - | - | $T_3$ | $88.51_{\pm 0.61}$ |
| 18 | $\mathbb{R}^3$ | ✓ | - | ✗ | - | - | none | $88.47_{\pm 0.31}$ |
| 19 | $\mathbb{R}^3$ | ✓ | - | ✓ | - | - | none | $\mathbf{89.02}_{\pm 0.07}$ |
| | | Reference methods from literature | | | | | | |
| PointNet | - | - | - | - | - | - | - | 90.7 |
| PointNet++ | - | - | - | - | - | - | - | 93.0 |
| DGCNN | - | - | - | - | - | - | - | 92.6 |

Table 7: Ablation study on ShapeNet 3D part segmentation reporting instance mean IoU for a randomly rotated dataset, comparing validation-set performance when trained on different percentages of the training dataset

| | | Rapidash with internal SE(3) Equivariance Constraint | | | | | | | | | |
|---|---|---|---|---|---|---|---|---|---|---|---|
| Model Variation | Type | Coordinates as Scalars ! | Coordinates as Vectors ! | Normals as Scalars ! | Normals as Vectors | Global Frame ! | Effective Equivariance | IoU (rotated) ↑ 100% | IoU↑ 80% | IoU↑ 60% | Normalized Epoch Time |
| ➡ 8 | $\mathbb{R}^3 \times S^2$ | ✗ | ✗ | ✗ | ✓ | ✓ | SE(3) | 85.45 | 85.46 | 85.10 | |
| 9 | $\mathbb{R}^3 \times S^2$ | ✗ | ✓ | ✗ | ✗ | ✗ | SO(3) | 84.48 | 84.21 | 84.07 | - |
| | | Rapidash with Internal $T_3$ Equivariance Constraint | | | | | | | | | |
| 17 | $\mathbb{R}^3$ | ✗ | - | ✓ | - | - | $T_3$ | 83.00 | 82.50 | 82.03 | - |

3. **SE(3) (Roto-translation Equivariant)**: Model utilizing the $\mathbb{R}^3 \times \mathbb{S}^2$ base space, where the kernel is conditioned on the three SE(3) invariant attributes, as introduced in [Bekkers et al., 2024], and given in Eq. 4.

### E.2.1  Molecular Generation: Performance and Efficiency

The primary goal of this experiment is to assess whether the performance gap observed in the main text (where SE(3) models are constrained by fewer parameters) can be closed by giving less constrained models a generous training budget and scaling up their capacity to match the computational complexity of the SE(3) models.

**Performance Analysis (Fig. 7)**   The results confirm that *equivariance is a critical inductive bias* for the molecular generation task. The unconstrained "None" models consistently failed to produce sensible molecules despite receiving ample training time and parameters, suggesting that a lack of geometric constraint leads to catastrophic performance for this task. The $T_3$ models performed remarkably well, even surpassing previous state-of-the-art results on QM9 generation. However, *peak performance is still decisively reached through the SE(3) inductive bias*. This suggests that even when capacity and training time are scaled liberally for the simpler $T_3$ model, the geometric constraint provided by SE(3) equivariance allows for significantly better generalization and performance.

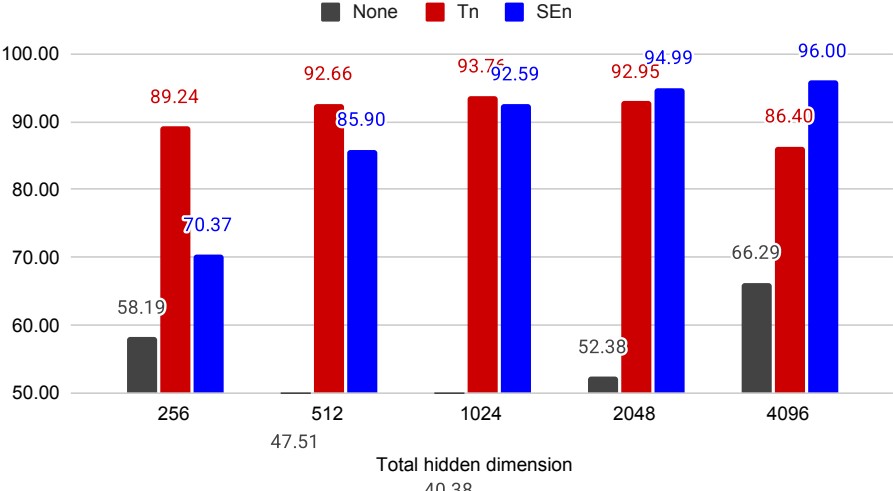

Figure 7: Comparison of generative performance (Molecule Stability %) on an equal training budget (48 hours, same GPU). All models leverage positional information. The labels "None", "$T_3$", and "SE(3)" denote equivariance to the trivial, translation, and roto-translation group, respectively.

This result shows that increasing model scale does not eliminate the performance advantage of a well-aligned SE(3) inductive bias.

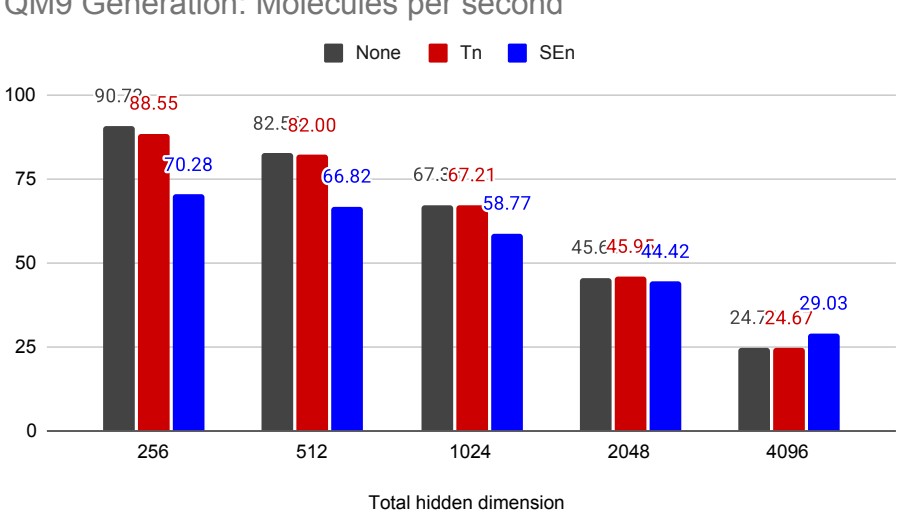

Figure 8: Comparison of test-time generation speed, reported as the number of molecules generated per second. Molecules were sampled in batches of 128 over 50 denoising steps.

**Efficiency Analysis (Fig. 8)** The generation speed linearly decreases as the hidden dimension increases across all models. For any given hidden dimension, the speed is approximately constant regardless of the inductive bias used (None, $T_3$, or SE(3)). This indicates that the SE(3) constraint does not impose a significant or prohibitive computational overhead in the inference phase of the generative model compared to the less constrained alternatives. At the highest hidden dimension

tested (4096), the SE(3) model is only marginally faster (approximately 17.7%), but its moderate efficiency is *vastly outweighed by the performance gain* over the other models.

### E.2.2 Molecular Property Prediction: Performance and Efficiency

For the property prediction task (regression), we use the Mean Absolute Error (MAE) on the $\mu$ property (dipole moment) of QM9.

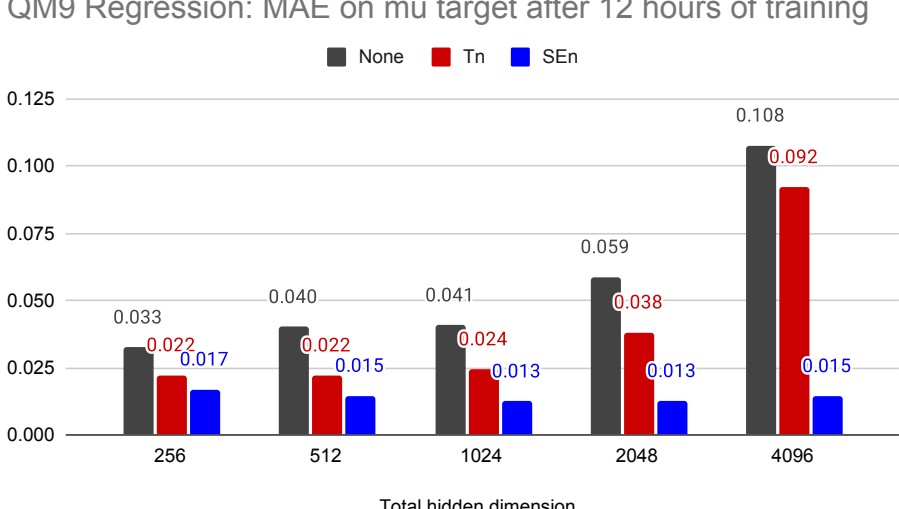

Figure 9: Comparison of regression performance (MAE on $\mu$) on an equal training budget (12 hours, same GPU). The labels "None", "$T_3$", and "SE(3)" denote the level of symmetry constraint.

**Performance Analysis (Fig. 9)** The performance hierarchy is clear and stable across all tested hidden dimensions: SE(3) $\gg T_3 \gg$ None.

- The SE(3) models are consistently the best performing and remain stable as capacity scales.
- The critical observation is that the increased network capacity in the "None" and $T_3$ models *cannot compensate for the lack of a strong geometric inductive bias*. In fact, their performance tends to degrade or remain stagnant with increasing hidden dimensions, suggesting they reach the limits of their geometrically defined hypothesis space and potentially struggle with optimization or overfitting in the more expressive space.

This strongly supports the argument that the *architectural choice of an appropriate inductive bias is more critical than simply scaling capacity*.

**Efficiency Analysis (Fig. 10)** Similar to the generative task, the prediction speed decreases linearly with the hidden dimension. The computational cost across all three models for a given hidden dimension is within the same order of magnitude. The SE(3) model is only marginally slower (approx. 19.8% at the lowest hidden dimension and 19.3% *faster* at the highest). Considering the significant performance gap (SE(3) achieving much lower MAE), the *moderate cost of the SE(3) inductive bias is justified by the large generalization gain*. It is safe to conclude that the efficiency differences do not render any model variant computationally prohibitive compared to the others.

## F  Implementation details

We implemented our models using PyTorch [Paszke et al., 2019], utilizing PyTorch-Geometric's message passing and graph operations modules [Fey and Lenssen, 2019], and employed Weights and Biases for experiment tracking and logging. A pool of GPUs, including A100, A6000, A5000, and 1080 Ti, was utilized as computational units. To ensure consistent performance across experiments, computation times were carefully calibrated, maintaining GPU homogeneity throughout.

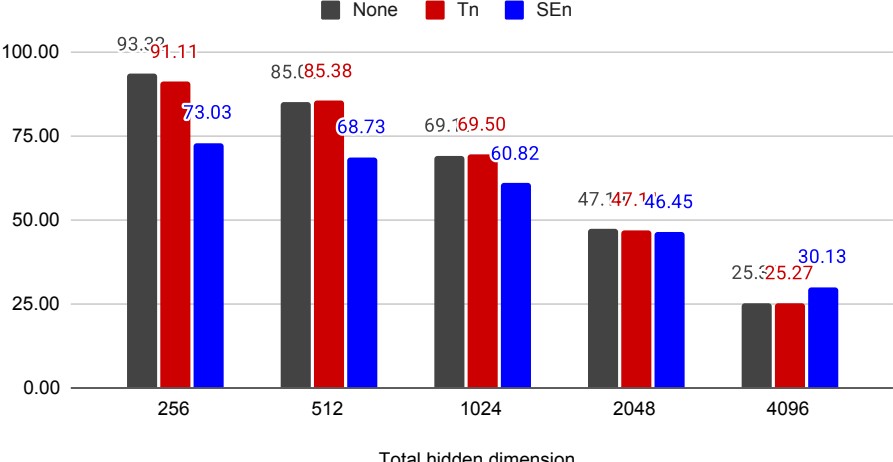

Figure 10: Comparison of test-time prediction speed, reported as the number of predictions (per sample) made per second. Predictions were made with a batch size of 128.

For all experiments, we use `Rapidash` with 7 layers with 0 fiber dimensions for $\mathbb{R}^3$ and 0 or 8 fiber dimensions for $\mathbb{R}^3 \times S^2$. The polynomial degree was set to 2. We used the Adam optimizer [Kingma and Ba, 2014], with a learning rate of $1e-4$, and with a CosineAnnealing learning rate schedule with a warm-up period of 20 epochs.

All models for equivariant tasks are trained with SO(3) augmentation. For ShapeNet segmentation, we use standard augmentation (20 degree z-rotations + scaling), as full SO(3) augmentation hurt in-distribution performance.

**QM9** QM9 dataset Ramakrishnan R. [2014] contains up to 9 heavy atoms and 29 atoms, including hydrogens. We use the train/val/test partitions introduced in Gilmer et al. [2017], which consists of 100K/18K/13K samples respectively for each partition.

**ShapeNet 3D** ShapeNet 3D dataset [Chang et al., 2015] is used for part segmentation and generation tasks. ShapeNet consists of 16,881 shapes from 16 categories. Each shape is annotated with up to six parts, totaling 50 parts. We use the point sampling of 2,048 points and the train/validation/test split from [Qi et al., 2017]. For this task, we trained we trained model variation (1-4 & 16-19) in Tab.5 and Tab.3 with two different settings of hidden features, C = 256 (gray) and C = 2048. The later inflated model was trained to match the representation capacity of the rest of the models. For the segmentation task, we use rotated samples and compute IoU with aligned and rotated samples. All the models were trained for 500 epochs with a learning rate of $5e-3$ and weight decay of $1e-8$.

For the segmentation task, we trained we trained model variation (1,2 & 6,7) in Tab.3 with two different settings of hidden features C = 256 (gray) and C = 2048. The latter inflated model was trained to match the representation capacity of the rest of the models. All the models were trained for 500 epochs with a learning rate of $5e-3$ and weight decay of $1e-8$.

**CMU Motion Prediction** For the motion prediction task, we evaluate our models on the CMU Human Motion Capture dataset [Gross and Shi, 2001], consisting of 31 equally connected nodes, each representing a specific position on the human body during walking. Given node positions at a random frame, the objective is to predict node positions after 30 timesteps. As per Huang et al. [2022] we use the data of the 35th human subject for the experiment. See Figure 11 to see instance of the dataset. For this task, we trained model variation (1-4 & 16-19) in Tab. 5 with two different settings of hidden features, C = 256 (gray) and C = 2048. The latter inflated model was trained to

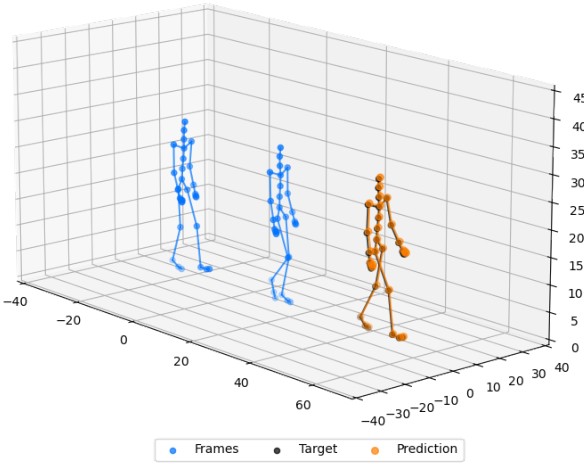

Figure 11: Depiction of an instance from CMU motion capture dataset.

match the representation capacity of the models (5-15). All the models were trained for 1000 epochs with a learning rate $5e - 3$ and weight decay of $1e - 8$.

**ModelNet40** For this task we trained 9 model variations presented in Tab. 6 All the models were trained for hidden features C = 256 for 120 epochs with a learning rate $1e - 4$ and weight decay of $1e - 8$.

**Diffusion Model** Unlike EDM Hoogeboom et al. [2022] that uses a DDPM-like diffusion model with a deterministic sampler, we use a stochastic sampler proposed in Karras et al. [2022]. We condition the diffusion model with feature scaling and noise scaling and combine outputs with skip connections, allowing for faster sampling. The sampler used in this work implements a stochastic differential equation with a second-order connection.

**Geometric task complexity** Classification/Regression does not change with group transformation of the input, thus invariant to the transformation. Segmentation with/without a global frame is an invariant/equivariant task. Dynamics (It varies with the actual dynamics (e.g. can be a simple motion equation vs fluid dynamics with instability terms and inputs (positions, velocity, additional parameters, thresholds etc), is generally an equivariant task, and molecular generation is an equivariant task, often with certain symmetry breaking to form different structures. Here is a figure depicting geometric task complexity for the tasks covered in the paper.

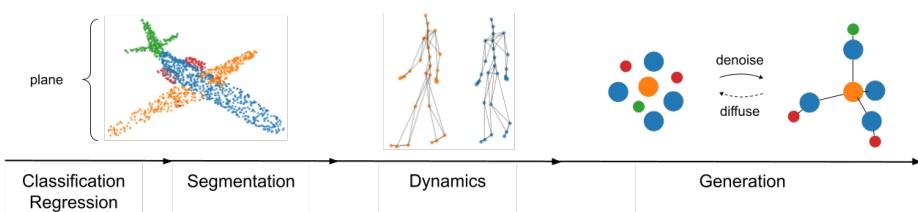

Figure 12: Progression of tasks based on geometric complexities of each task.

