# OpenReview forum: "Probing Equivariance and Symmetry Breaking in Convolutional Networks"
_NeurIPS.cc/2025/Conference — NeurIPS 2025 poster_

### Official Review · Reviewer_V9ah · 2025-07-01

**Clarity:** 2
**Significance:** 2
**Originality:** 2
**Rating:** 3
**Confidence:** 4

**Summary:**

This paper introduces an architecture with flexible symmetry parametrisation based on regular group convolutions on R^3xS_2.

The paper then investigates incorporating various levels of symmetry when solving a number of tasks that may benefit from it.

The paper draws a number of conclusions about the importance of symmetry in this architecture, data scaling, model capacity scaling, and symmetry breaking.

**Questions:**

1. Do the authors think that adding data augmentation to the non-equivariant or reduced equivariant architectures would make a difference to the results? If so, should this be included in the ablation studies?
2. Do the author expect these conclusion to hold across other types of equivariant architectures? Open source code exist for a number of other equivariant architectures, e.g. https://github.com/QUVA-Lab/escnn, https://github.com/vgsatorras/egnn, which could easily be adapted to the reduced symmetry and symmetry breaking settings of the paper, should comparisons to these alterative architeture types be included to make the conclusions of the paper more substantial?

**Ethical Concerns:**

["NO or VERY MINOR ethics concerns only"]

**Limitations:**

Yes

**Quality:**

2

**Strengths And Weaknesses:**

Strengths:
- The evaluation if reasonably well conceived, although the mix of tasks could be wider to disambiguate effects over multiple datasets to disentangle and dataset specific effects.
- The layout of hypotheses are clear

Weaknesses:
- The authors seem to miss a key question regarding equivariant models in their analysis, eluded to in the introduction. Namely, that when equivariance is removed  from an architecture it is typically replaced with data augmentation. Without this inclusion I find the conclusions of  RQ1-3 tricky to conclude properly.
- The authors only look at a single architecture. The conclusions seem unlikely to hold across all classes of model to me. In particular, with irregular representation convolutions vs regular representation convolutions, the additional complexity in parametrisation and runtime cost seem likely to significantly change the results of the expreiments.

---

> ### Author Rebuttal · Authors · 2025-07-31
>
> Dear reviewer, we sincerely thank you for the time and effort you put into providing feedback on our work. We deeply appreciate your insightful questions and respond to the concerns and questions below:
>
> ## Data augmentation [W1]
>
> Thank you for pointing out that we do not emphasize in the main paper that the experiments are performed with augmentations.  We will update the main paper with details on augmentations. **All models for equivariant tasks are trained with SO(3) augmentation**. For ShapeNet segmentation, we use standard augmentation (20° z-rotations + scaling), as full SO(3) augmentation hurts in-distribution performance. In the code added as supplementary material, we have config files with the augmentation parameter (set as true) for equivariant tasks, which can be used to easily reproduce the results in the paper.
>
> ## Alternate architectures [W2]
>
> Generally, for this work, we refrain from further comparison with additional model classes, as we believe our findings about equivariance effects should, *in principle, generalize across model classes since our ablations isolate **fundamental properties of *equivariance constraints***.  Only within a single architectural class can we do proper ablation studies, and thus, we can attribute performance gains/drops to changes in equivariance constraints. If we want to explore whether or not the results generalize to other architectural classes, we have to develop such a new class of models in addition to our already proposed flexible architecture (Rapidash) and redo the entire study (which includes **58 model variations across 5 tasks**). We consider this to be beyond the scope of the current paper and decide to emphasize that these results hold within the convolutional neural network class (both in $\mathbb{R}^3$ and $\mathbb{R}^3 \times S^2$) paradigm. Related works like [1], answer a few questions about the role of equivariance with a transformer-based model and share similar trends as our findings in scaling models for equivariant tasks.
>
>
>
> ## Response to Q1: Augmentations
>
> Yes, the **presented results are with augmentations**.  We will add results without augmentations in the Appendix in our updated version. Also see W1.
>
> ## Response to Q2: Alternate architectures
>
> Related works like [1] answer a few questions about the role of equivariance with a transformer-based analysis and share **similar trends** as our results in scaling models for equivariant tasks. We believe the trends of our findings hold true across the mentioned architectures, ESCNN and EGNN, although it can be said with certainty only with a **detailed analysis** of each model class. Having extensive analysis for each available equivariant architecture/class would be very useful for the community.  We leave that as future work.
>
> We once again thank you for taking the time to review our work. We believe the results in this paper offer relevant insights for both theory and practice *in convolutional architectures*, and we would appreciate your reconsideration of the evaluation in light of these clarifications.  If any concerns remain from your initial evaluation, we’d be happy to clarify further.
>
> Reference:
>
> [1] *Does equivariance matter at scale?*, Brehmer et al.

---

> > ### Comment · Reviewer_V9ah · 2025-08-08
> > **Reply to rebuttal**
> >
> > Thank you to the Authors for engaging in the reveiw.
> >
> > Q1: Thank you for clarifying this. I think it is important to include the results of both the unaugmented and augmented runs of the models without equivaraince. I also note the authors train with SO(3) augmentation, but not any T3 augmentation even on tasks where this might be relevant. Being able to disentangle the performance gains over baseline models is important, from both augmentation and equivariance.
> >
> > Q2: In light of the clarification, I think the claims of the paper and the title are overreaching then. This paper looks at regular G-convolutions on R^3xS_2 at most, and explores SE(3) equivariance only. The tasks explored are point cloud tasks only. The claim to investigate equivariance in all convolutional networks seems to great to me.
> >
> > Additional points:
> > - The authors do not perform an extensive capacity study of the models in the paper, which can have a significant effect on the outcome of results - e.g. see Figure 4 in [1].
> > - It still remains unclear to me the exact contributions of this paper. The architecture proposed is mostly derived form prior work. There are then 5 hypotheses investigated in a short amount of space, without a significant common thread between these hypotheses beyond equivariance, and in my opinion with not enough data to draw broad conclusions that the paper claims. I believe the paper would be better suited to focus on a smaller number of related questions, or take the time and space to investigate each of the questions properly, such as [1].
> >
> > [1] Does equivariance matter at scale?, Brehmer et al.

---

> ### Author Response · Authors · 2025-08-05
> **Kind Invitation for Further Discussion**
>
> Dear reviewer,
>
> We thank you again for a detailed and constructive review of our work. We would like to kindly invite you to take a look at our rebuttal, in which we have addressed each of your comments thoroughly. In particular:
> - Added **data-augmentation** in our discussion, as well as clarified that the experiments have data-augmentation.
> - Motivated our focus on a **single architecture class** for this work and added discussion about additional architectures.
>
> We think your feedback has helped us improve our work, and we thank you again for your valuable time and feedback!
> Kindly let us know if you have any further questions or concerns.

---

> ### Comment · Area_Chair_q2ou · 2025-08-07
>
> Dear reviewer, could you please spend some time discussing with the authors or commenting on their responses? This paper receives diverging scores, and it would be helpful to have more discussions. To note, in this year's NeurIPS, it is required to have reviewer-author discussions before submitting the mandatory acknowledgement. Thanks!

---

> ### Author Response · Authors · 2025-08-08
> **Response to reviewer V9ah**
>
> Dear reviewer,
>
> Thank you for your thoughtful engagement with our work. We appreciate your feedback and address your concerns below.
>
> > Q1
>
> We will add experiments for non-equivariant models without augmentations. Regarding T3 augmentations, we did not see them necessary for QM9 datasets, as we work on the Center of Mass frame for the entire task. Regarding CMU motion and ShapeNet, we agree with the reviewer and will add experiments with T3 augmentations in addition to the SO(3).
>
> > Q2
>
> We agree with the reviewer on the focus of the paper on point clouds ( also see our response to reviewer KUZW. In the updated version of the paper, we will make sure the **scope of the paper is clear** both in the introduction as well as the abstract.
> Regarding the point about G-convolutions, we thoroughly investigate SE(3) convolutions both in $\mathbb{R} ^3$ as well as $\mathbb{R}^3 \times S^2$ across various datasets and tasks. Additionally, we also provide a flexible architecture
> that allows for fair comparisons across different equivariance as well as symmetry breaking.
>
> >Additional points:
> The authors do not perform an extensive capacity study of the models in the paper, which can have a significant effect on the outcome of results - e.g. see Figure 4 in [1].
>
> Could you please let us know **what additional experiments** you propose? It would be great to add it to the updated version.
>
> > It still remains unclear to me what the exact contributions of this paper are. The architecture proposed is mostly derived form prior work. There are then 5 hypotheses investigated in a short amount of space, without a significant common thread between these hypotheses beyond equivariance, and in my opinion with not enough data to draw broad conclusions that the paper claims. I believe the paper would be better suited to focus on a smaller number of related questions, or take the time and space to investigate each of the questions properly, such as [1].
> [1] Does equivariance matter at scale?, Brehmer et al.
>
> The main contributions of the paper are the **empirical study** answering important research questions to understand the utility of the equivariance, a **theoretical formulation** to understand symmetry breaking through additional information, as well as **a flexible architecture** that allows for comparison across different equivariance and symmetry breaking. This is not possible in the prior work [1] as it is limited to fixed SE(3)/T3 equivariance and cannot handle larger point clouds.
>
> We would like to politely point out to the reviewer that while [2] is a great contribution in understanding the role of equivariance in scaling data and scaling models, as the title suggests, they do not consider other important aspects of model design while asking this question. They also present limited experiments and a lack of variety of tasks.
>
> We are working towards **adding a larger dataset**, Cosmobench [3], as suggested by reviewer KUZW, which we believe to be beneficial in extending the insights to large point clouds.
>
>
> We would like to thank the reviewer again for the useful feedback, which has helped us improve our work. We thank you again for your valuable time. Kindly let us know if you have any further questions or concerns.
>
> References:
>
> [1] *Fast, Expressive SE3 Equivariant Networks through Weight-Sharing in Position-Orientation Space*, Bekkers et al
>
> [2] *Does equivariance matter at scale?*, Brehmer et al.
>
> [3] *CosmoBench: A Multiscale, Multiview, Multitask Cosmology Benchmark for Geometric Deep Learning*, Huang et al.

---

### Official Review · Reviewer_KUZW · 2025-07-02

**Clarity:** 3
**Significance:** 2
**Originality:** 2
**Rating:** 4
**Confidence:** 4

**Summary:**

While equivariant kernels ensure generalization to specified groups, this comes at the cost of potentially over constrained networks on imperfectly symmetric datasets, generally lower network capacity, and higher compute requirements for a fixed parameter count. This has led to several papers that investigate the utility of equivariance by design on disparate tasks and datasets, some of which find benefits, whereas others do not. As a result, the literature currently conflicts with itself in terms of relevance to real world datasets and a unified analysis is needed.

This submission specifically pertains to deep learning on point clouds and provides a group-equivariant framework that enables isolating various key decisions (e.g., input feature types, equivariance constraints, etc.) such that they can be studied piece-by-piece. The paper also presents some new theoretical results to clarify when and how symmetry breaking and equivariance are useful depending on the task. Lastly, experimentally, it covers a broad set of tasks and domains on toy small-scale datasets such as ShapeNet, QM9, etc.

**Questions:**

Please correct me if I misunderstood any central points and I would be happy to reconsider my rating. Currently, I would very likely raise my score if:
- The paper clearly revises itself to narrow its scope to point cloud analysis and to describe which of its findings have the potential to translate across fields.
- The empirical investigations are expanded to include at least one large-scale real-world dataset with *well tuned* baselines used as a point of comparison (with extensive augmentation and/or regularization). It is understandable if this is not doable within the time allocated to the rebuttal, but in that case, please provide a rigorous discussion regarding the choice of datasets and what we can extrapolate to real data.
- It is clearly disambiguated which of the theoretical results are truly specific to this paper and which are inherited from previous work.

**Ethical Concerns:**

["NO or VERY MINOR ethics concerns only"]

**Final Justification:**

The rebuttal addressed many of the concerns raised in my original review and therefore I lean towards accepting the paper. My primary remaining reservations correspond to some of the unaddressed topics, including:
1. Making broad statements regarding the practical utility of equivariant networks on point clouds based on empirical results from small-scale (and some synthetic) datasets alone.
2. The paper requires a significant revision to reframe itself as specific to point cloud analysis and better tying the research questions together.

However, the authors have committed to addressing 1 and 2 in the final revision. On net balance, the paper is a useful data point in this long-running debate regarding inductive biases vs. scale for atleast the specific domain of point clouds.

**Limitations:**

The discussion section includes 2 sentences on being compute bound in terms of running larger scale experiments and mentions limiting analysis to convolutional architectures. While I don't think that these are the primary limitations of this work, these are thoughtful inclusions that should be appreciated.

**Quality:**

3

**Strengths And Weaknesses:**

## Strengths:
- The problem space of "*does equivariance by design matter*" is gigantic in scope and this paper does an admirable job in trying to construct a unifying framework for empirical investigation across a wide set of tasks.
- Section 2, background is very well done and accessible to a broad audience.
- New theoretical results on when breaking equivariance and symmetry help in generalization (and when they do not).

## Weaknesses:
### Over-claimed scope and toy experiments:
The paper claims more within its scope than it actually does. The paper primarily pertains to point clouds in all of its empirical investigations (which are its core contribution), yet the framing is left as general-purpose as possible, which IMO can be misleading. As an example, [Gruver&Finzi](https://openreview.net/forum?id=JL7Va5Vy15J) show that many of the statements made here do not translate to large-scale natural image-based vision where large networks without inductive bias trained on large datasets with extensive augmentation are practically *more equivariant* than equivariant-by-design architectures. This paper should revise itself to clarify its scope and which of its findings have the potential to translate across fields.

More broadly, the paper does not get to the true tension of the field: if equivariant architectures worked at scale across tasks, they would be used at scale across tasks by the community. There are some theoretical results provided, but all of the experiments are performed on simple toy datasets such as ShapeNet and QM9 wherein it is very easy to show great results that then do not translate to real tasks. I believe that the theoretical results should be validated on at least one large-scale real-world dataset with *well tuned* baselines used as a point of comparison.

### Unclear contributions:
It is hard to say definitively what to take away from the paper. The theoretical framework, Rapidash, is a multiscale extension of Ponita in [Bekkers, 2024](https://arxiv.org/pdf/2310.02970) (I suppose this naming scheme makes sense if you have a pokedex :), but this is never explicitly stated in the paper and it is claimed as an entirely new framework. However, many of its theoretical properties are inherited from Ponita. As a result, a significant portion of the methods section recaps properties that are originally specific to Ponita (e.g., Sec 3.1) and it is hard to isolate what is truly specific to Rapidash. Also, this raises the burden of empirical completeness to make a strong contribution, in my opinion.

### Missing analysis of augmentation and regularization:
The paper's analysis misses key components of performance such as augmentation types and strengths and regularization via techniques such as weight decay. Augmentation is not mentioned within the paper and in practice is key to networks learning equivariance in practice ([Gruver&Finzi](https://openreview.net/forum?id=JL7Va5Vy15J)). It is plausible that the corresponding non-equivariant networks require significantly more regularization and/or augmentation strength (with more training steps) to lower-the-margin-to or even exceed their equivariant counterparts on these toy point cloud datasets.

Please detail the types, strengths, and probabilities of the augmentations and regularizations used, and whether they were held identical for both equivariant and non-equivariant networks. If they were, please perform a few experiments where the non-equivariant networks are well tuned.

### Unclear relationship between the research questions
The research questions posed in the methods section come off as a bag of loosely-related topics that are of independent interest within the field, each of which could be its own paper. For example, the transition into Section 3.2 is very abrupt as it goes into detail about a paper by Lawrence et al 2025 proposing a probabilistic interpretation of symmetry breaking that is (a) previously unintroduced (b) hard to parse and translate to the context of this paper without being very familiar with Lawrence, et al.

In the revision, please make explicit what ties these research questions together and provide necessary background in Section 2 instead of in the Methods, which should primarily cover what is new and/or specific to this paper.

### Presentation:
- The actual methodological contribution (Rapidash) is only described in the appendix, but the main text of the paper writes at length about its properties. It is hard to follow if you don't know what the method actually is. The description should be moved into the main text and a lot of space can be made by removing unnecessary portions including L65--68 and L233--245 (which just restates the intro paragraph).
- The paper's opening paragraph doesn't contextualize the state of the field well. It comes off as just listing papers without outlining cohesive themes across papers. The latter would be much more accessible to readers who are not following the monthly back-and-forths on this contentious topic.

## Minor comments that do not impact my rating:
- Related work of interest: In the interest of unification, Gruver & Finzi 2023 should absolutely be discussed in this paper and how its findings on images could relate to the findings on point clouds here. Not essential but potentially interesting in the context of discussing Rapidash, there are works on position-orientation deep learning in medical imaging (e.g., in vessel analysis and diffusion MRI) that could be worth briefly discussing for better contextualization.
- L20: Please clarify that it's AlphaFold3 and not other versions of AF that did use equivariant architectures.
- Fig2: The radar plot + seemingly random text snippets make the figure very hard to parse. Please split into two separate subfigures.
- L218: Weight sharing in CNNs was introduced quite a bit earlier than LeCun2010 :)
- The connections made to tensor field networks can be left at a sentence with the rest moved to the appendix. There's already too much packed into the main text of the paper as is.

---

> ### Author Rebuttal · Authors · 2025-07-31
>
> Dear reviewer,
>
> We thank you for the time and effort you have put into providing a review of our work. We sincerely appreciate your thoughtful feedback. We are grateful that you find our work an **admirable job** in constructing a unifying framework. We appreciate your kind words about the background as well as the **theoretical results**.
>
> In the text below, we try to address your concerns. We appreciate the careful suggestions that have helped improve our work and will make the changes in the updated version.
>
> ## Scope of the paper
>
> - We agree with the reviewer that the focus of **our work is on point clouds**, and we will **emphasize** this in the introduction as well as in the contribution list. We will revise the wording in the paper to emphasize the point clouds aspect. Thank you for pointing us to [1], and will add this to the related work/discussion to share how some insights do not translate across fields, i.e, images to point clouds.
> - In the scope of this paper, we *do not* tackle **regularization**. We will add this to our limitations and consider this in future works to understand the role of regularization in equivariance learning.
> - We would like to point out that Rapidash architecture, unlike PONITA [1], allows for processing a larger number of points per sample. Due to various **different input options**, Rapidash is a flexible architecture that allows for detailed empirical analysis of different group equivariance as well as symmetry/equivariance breaking. In the updated version of the paper, we will add how Rapidash differs from [1] explicitly.
>
> ## Connecting research questions presented in the paper
>
> The **five core research questions** in our work focus on kernel constraints, representation capacity, dataset size, symmetry, and equivariance breaking. We think these are crucial when it comes to understanding the role of equivariance.
>
> - RQ1 addresses the role of constraining the kernel and where that helps or hurts model performance.
>
> - RQ2 and RQ3 help understand if increased model size or increased dataset size can help learn equivariance.
>
> - RQ4 and RQ5 connect to understand how to incorporate more information in data, especially if that leads to symmetry breaking or equivariance breaking, and which of these can be useful for better performance.
>
> ## Presentation
>
> - We fully agree with the reviewer that the detail of Rapidash should be in the main text. Due to lack of space, the **details of the Rapidash architecture** are only discussed in the Appendix. We will incorporate the suggestion and add it to the main text of the paper.
> - We will **rewrite** parts of the introduction to better describe the related works in a way that is more focused on the theme of the work.
> - We will also provide the **necessary background** in Section 2 to help understand both the theoretical contribution and research questions cohesively.
>
> ## Clarification on theoretical contributions
>
> Proposition 3.2 [main text], Corollary 3.1 [main text], Proposition D.1 [Appendix], and Proposition D.2 [Appendix] are the theoretical contributions in this work, while the rest of the propositions/corollaries are presented from earlier works for completeness.
>
> ## Choice of datasets
>
> With the focus of the paper being on point clouds, we picked the most commonly benchmarked small molecule dataset **QM9** (with **generation/property prediction** task). For adding a different task, learning dynamics, we picked **CMU Motion** dataset. Given that it is a very small dataset, we wanted to understand the role of equivariance in a small data regime. For the rest, we pick the commonly used standard pointcloud datasets, i.e., **ShapeNet3D** (for **segmentation and generation** tasks) and **ModelNet40** for **classification**. These datasets collectively give a **good range** of tasks along with additional inputs *(i.e. global pose, global frame, etc )* allowing for different types of symmetry/equivariance breaking on point cloud datasets of varied size.
>
>
>
> ## Regarding large-scale datasets
>
> In response to the point raised regarding large-scale datasets, we are working towards including the **COSMO-bench** dataset [1] (comprising of 5000 point clouds, *Quijote, coarse-grained* per sample as well as a physics baseline in addition to a GNN baseline for velocity prediction tasks for ease of comparison). We are working on having *Rapidash + variants* for prediction tasks and will add the results in the updated version of the paper. If the reviewer has other suggestions, kindly let us know.
>
> ## Response to minor comments
>
> - We will **update our discussion** to include [2] as well as other works in the position-orientation space.
> - We will update the AlphaFold version in the introduction and weight-sharing **citation**.
> - With regards to the figure, we will add the **split version** of Figure 2 to the Appendix.
>
>
> We once again thank you for taking the time to review our work, and we would appreciate your reconsideration of the evaluation in light of these clarifications.  If any concerns remain from your initial evaluation, we’d be happy to clarify further.
>
> References:
>
> [1] *Fast, Expressive SE(n) Equivariant Networks through Weight-Sharing in Position-Orientation Space*, Bekkers et al.
>
> [2] *The Lie Derivative for Measuring Learned Equivariance*, Gruver & Finzi et al.
>
> [3] *CosmoBench: A Multiscale, Multiview, Multitask Cosmology Benchmark for Geometric Deep Learning*, Huang et al.

---

> > ### Comment · Reviewer_KUZW · 2025-08-05
> >
> > Thank you for the rebuttal -- I have read and considered it.
> >
> > I'd like to discuss these a few of these topics further before finalizing a rating. For context, none of these are standalone *must-haves*, but I want to better understand which of the paper's claims and experiments can translate into actual practice.
> >
> > #### Scope of the paper:
> > - Regarding "*(we) will add this to the related work/discussion to share how some insights do not translate across fields, i.e, images to point clouds.*": If the paper is now specific to point clouds and doesn't transfer (to images, e.g.), this would conflict with the introduction's claims (e.g., paragraph 1) that this work unifies the self-contradictory literature on the benefits of equivariance at scale across applications. IMO, the paper's title also needs to change to acknowledge its more restricted scope of just point clouds.
> > - Regarding not tackling regularization: Was regularization not used for the paper's baselines? If not, why? In practice, regularization (e.g., consistency losses or just simply tuning the weight decay) can narrow the gap between models and seems relevant.
> >
> > #### Connecting research questions:
> > I think the point in the review might have been missed. I understand the research questions, I want to better understand what ties these questions together. For context,
> > - IMO, all of these questions could constitute independent papers with deeper analyses instead of the (necessary due to the conference format) more superficial analyses presented here.
> > - The example in the original review about the abrupt transition into Lawrence et al 2025 in the original review requiring substantial background and familiarity with that specific work was not engaged with.
> >
> > If the takeaway connection between the topics is "*five questions equivariance practitioners talk about*", then one could reasonably ask why it excludes other questions of practical relevance (e.g. regularization, data efficiency, etc.). The paper would be stronger if it had a concrete connecting theme between its questions, such that a unified background and takeaway could be presented.
> >
> > #### Large-scale datasets:
> > I appreciate the proposed experiments on COSMO-bench, but I don't know what to make of this argument if the results on that dataset aren't presented in the discussion period. The eventual results may or may not support the paper's arguments.
> >
> > #### Misc:
> > From the followup message: "*(we) answered your concerns regarding **augmentations***": I don't see a response to the missing-augmentation-analysis point in the rebuttal. Am I missing something?

---

> ### Author Response · Authors · 2025-08-05
> **Kind Invitation for Further Discussion**
>
> Dear reviewer,
>
> We thank you again for a detailed and constructive review of our work. We deeply appreciate the time and expertise you invested in evaluating our work.
> We would like to kindly invite you to take a look at our rebuttal, in which we have addressed each of your comments thoroughly. In particular:
> - Re-defined the **scope** of our work to focus on point clouds.
> - Clarified our **theoretical contributions**.
> - **Connected** the presented **research questions**.
> - We are currently working towards adding a **large-scale dataset** and will add those results in the Appendix.
> - Answered your concerns regarding **augmentations** as well as made changes to the presentation of our work.
>
> We think your feedback has helped us improve our work, and we thank you again for your valuable time and feedback!
> Kindly let us know if you have any further questions or concerns.

---

> ### Author Response · Authors · 2025-08-07
>
> Dear reviewer,
>
> Thank you for your thoughtful engagement with our work. We appreciate your feedback and address your concerns below.
>
> >Scope of the paper:
> If the paper is now specific to point clouds and doesn't transfer...
>
> We will update the paper's abstract as well as introduction (as previously mentioned) to clarify the scope of the paper to point clouds. (Not sure if we are allowed to change the title after submission, regardless, we will ensure this comes through.) Regarding unifying self-contradictory literature, our goal is not to contrast equivariant and non-equivariant models in isolation, but also to consider relevant symmetry breaking and the task at hand before making a selection; for this, the presented architecture will serve as a useful tool. With regards to images, for instance, we would like to emphasize [2], which highlights the lack of genuine invariance learning while also focusing on true local equivariance loss in various models as given in [1]. We would like to highlight that there is still much to uncover about the inductive biases of model classes like transformers, and that could play a role in the results presented in [1, 5].
>
> >Regarding not tackling regularization..
>
> While responding to your questions about regularization, we particularly meant explicit regularization loss like that in [3,4], which we have not explored. To best of our knowledge, there is no general equivariant regularization loss (*if there is one, and we have missed that, please do suggest so we can add this to our model in the future*), so we focused only on weight decay. We tune weight decay as part of the hyperparameter optimization, and we did not find a significant impact on either model class (equivariant or not). For our experiments, we have weight decay as 1e-8 for QM9 and 1e-12 for the rest. We observe that weight decay does not compensate for a potential loss of equivariance.
>
> > IMO, all of these questions could constitute independent papers with deeper analyses instead of the (necessary due to the conference format) more superficial analyses....
>
> We politely disagree with the reviewer on our work as *superficial analysis*. We think this work provides useful insights about equivariance and symmetry breaking for point clouds while giving a unified architecture that allows for a fair comparison. As noted in the rebuttal, we will update our background section to have a better flow, including a clearer introduction for Lawrence et al. We consider the proposed research questions under the broader question about the utility of equivariance (in models for point clouds), and while doing so, we capture aspects of building equivariant models through kernel constraints, strictness of equivariance, as well as symmetry-breaking aspects.  We’ll also provide a more cohesive overview of relevant work on equivariance learning. As suggested by reviewer mQdo, we will add explicit takeaways (see our response to mQdo).
>
> > Large-scale datasets:
> I appreciate the proposed experiments on COSMO-bench, but I don't know what to make of this argument if the results on that dataset aren't presented in the discussion period...
>
> If the results of large-scale datasets do not show the same trends as the datasets presented in the paper, we believe that this will give us insight into how learning equivariance can be dependent on the scale of the dataset. So either way, we believe it would be useful. From our preliminary results, we see that the equivariant model outperforms the given (non-equivariant) baseline from the Cosmo-bench [6], and we hope to include all the Rapidash variants in the final version of the paper, as this is not possible in the short discussion period.
>
> >Misc:
>
> Apologies for not highlighting our response to reviewer mQdo previously.
>
> All models for equivariant tasks are trained with SO(3) augmentation. For ShapeNet segmentation, we use standard augmentation (20° z-rotations + scaling), as full SO(3) augmentation hurts in-distribution performance. In the code added as supplementary material, we have config files with the augmentation parameter (set as true) for equivariant tasks, which can be used to easily reproduce the results in the paper.
>
> We thank you again for your valuable time and feedback! Kindly let us know if you have any further questions or concerns.
>
> References:
>
> [1] *The Lie Derivative for Measuring Learned Equivariance*, Gruver and Finzi et al.
>
> [2] *On genuine invariance learning without weight-tying*, Moskalev et al.
>
> [3] *Latent Space Symmetry Discovery*, Yang et al.
>
> [4] *Improving Transformation Invariance in Contrastive Representation Learning*, Foster et al.
>
> [5] *The Importance of Being Scalable: Improving the Speed and Accuracy of Neural Network Interatomic Potentials Across Chemical Domains*, Qu et al.
>
> [6] *CosmoBench: A Multiscale, Multiview, Multitask Cosmology Benchmark for Geometric Deep Learning*, Huang et al.

---

### Official Review · Reviewer_mQdo · 2025-07-03

**Clarity:** 3
**Significance:** 3
**Originality:** 3
**Rating:** 5
**Confidence:** 3

**Summary:**

The paper introduces a unified architecture, Rapidash, which unifies non-equivariant and equivariant models, and assess the effect of models with different structural priors, including forms of symmetry/equivariance breaking, to different tasks.

**Questions:**

Symmetries reduce the hypothesis space to functions that respect the symmetry (hence potentially limiting train fit capacity), but allow better generalization (test data fit) if the symmetries match the underlying data distribution, with symmetry breaking/relaxations being something in between.

Q1: Given the relevance of distinguishing train/test data, would it be possible to report both train and test loss. Currently it is not always clear whether MSE is on train or test data (assuming it’s the test metric for CMU), but it would be helpful if both can be provided? Adding these as an Appendix should be enough.

Q2: Related to this, results of the paper are particularly strong because the representational capacity is taken into account, ensuring that symmetric models do not merely generalize better due to undercapacity. The paper mentions that saturating performance suggests that ‘the limits of their architecturally defined hypothesis space for geometric expressivity’. Although true, it seems this could be easily verified by checking if models achieve low train error.

**Ethical Concerns:**

["NO or VERY MINOR ethics concerns only"]

**Final Justification:**

I am satisfied with the answers by the authors and maintain my recommendation for acceptance.

**Limitations:**

- The paper is very well-written. The main point of critique is the limited assessment of the train/test error difference, which seems highly relevant given that the paper focuses on measuring generalization capabilities in the context of structural geometric priors.
- Secondly, the evaluation and interpretation of the results could be clearer and more precise. The paper presents results in large tables, which would benefit from additional exposition and discussion.

**Paper Formatting Concerns:**

Formatting seems fine.

**Quality:**

3

**Strengths And Weaknesses:**

Strengths:
- The paper is well-written, provides an extensive study that ranges from theoretical analysis, empirical analysis, to practical architecture and empirical results on well-known data sets. The scope of this evaluation strengthens conclusions that are drawn.
- The paper builds upon a solid geometric framework established in prior work.
- Proposes a concrete architecture, Rapidash, that is useful to better understand and empirically evaluate geometric priors in the context of deep learning.
- Insightful results and findings that benefit the community to better understand how models with structural geometric priors can benefit practical applications. In particular, the work offers important insights in the effect of incorporating symmetry breaking or relaxations have on predictive performance

Weaknesses:
- Although the analysis is quite extensive, one minor point of critique is that it does not seem to consider the difference between training and test error as particularly relevant. However, this distinction is crucial for understanding the difference between fitting (symmetry-restricted hypotheses) and the ability to generalize (having the right inductive bias).
- The large result tables can be daunting at times and might benefit from clearer explanations in the text. In particular, it would be helpful to explicitly state the expected generalization behavior and explain how the results align with, or deviate from, those expectations. I recognize this may be a matter of personal preference and could also be constrained by page limits, so I won’t weigh it heavily in my overall assessment.

---

> ### Author Rebuttal · Authors · 2025-07-31
>
> Dear reviewer, we thank you for the time and effort you have put into providing a review of our work.
>
> Firstly, we sincerely thank you for your kind words and appreciation of our work. We are grateful that you found the paper to be **well-written** and recognized the extensive study spanning **theoretical analysis, empirical evaluation, and practical architecture** . We especially appreciate your recognition of our proposed architecture, Rapidash, as a *concrete and useful* contribution for understanding geometric priors in deep learning. We hope the community finds our work *insightful* as well.
>
> In the text below, we address questions and weaknesses/limitations.
>
> ## Train error  vs test error [W1, L1]
>
> We would like to mention that for all tasks, we train with CosineAnnealing and select the best checkpoint based on validation performance. For ShapeNet, which lacks a standard validation set, we use the final model; no overfitting was observed. As we consider the validation error to decide the best model, we did not report the val (nor train) error, but only the error on the designated test set.
>
> ## Readability of large tables/ Takeaways [W2, L2]
>
> We appreciate your feedback regarding the readability of large tables. In the revised version, we will:
>
> - Add **split** (smaller) versions of the large tables in the Appendix for improved readability.
>
> - Include clearer **takeaway messages** *(as shown below)* in the discussion section to enhance accessibility:
>   * The performance gap between equivariant and non-equivariant models increases with geometric task complexity.
>   *For simpler tasks like classification and regression, *using* non-equivariant models performs somewhat similarly to equivariant models.
>   * Symmetry breaking, for example, adding positional information, can be useful and give a performance boost.
>   * For equivariant tasks like molecular generation, using equivariant models is important, as non-equivariant models perform poorly.
>
> ## Response to Q1
>
> For the CMU dataset, we report **test MSE loss** and use this to compare models. We will add a table with train and test errors for the CMU dataset in the Appendix in the updated version.
>
> ## Response to Q2
>
> We mention ‘*the limits of their architecturally defined hypothesis space for geometric expressivity’* with respect to Table 3 for inflated models vs regular models segmentation task on ShapeNet (both rotated, non-rotated samples). We indeed see that making the models larger *does not improve* test error and will report that in the updated version of the paper.
>
> Once again, we thank the reviewer for the insightful feedback. If any concerns remain from your initial evaluation, we’d be happy to clarify further.

---

> ### Comment · Area_Chair_q2ou · 2025-08-07
>
> Dear reviewer mQdo, thanks for your time reviewing this paper. Although your opinion might be clear, according to this year's NeurIPS policy, it is required to have reviewer-author discussions before submitting the mandatory acknowledgement. Could you please take some time to comment on the authors' responses? Thanks!

---

### Official Review · Reviewer_Bpqw · 2025-07-05

**Clarity:** 3
**Significance:** 3
**Originality:** 3
**Rating:** 5
**Confidence:** 4

**Summary:**

This paper investigates the tradeoffs of utilizing various types of equivariant and unconstrained models for learning a broad range of 3D tasks. Due to the nature of the 3D tasks, the authors focus on SE(3) symmetries and introduce a scalable group convolutional architecture (called Rapidash) that allows them to control the level of equivariance and symmetry breaking applied to the models in a unified way. Using Rapidash, the paper investigates how different levels of equivariant constraints affect the performance of the models, how the size of the models impacts the various equivariant architectures, and how the performance of these architectures scales with different dataset sizes. Additionally, the authors investigate how applying a symmetry-breaking mechanism can affect the performance of the models. After defining the specific questions that they aim to answer, they perform an extensive evaluation of the different options on various 3D tasks and provide an in-depth discussion interpreting their empirical observations.

**Questions:**

- What is the role of augmentations when investigating the benefit of using equivariant models? Are augmentations and equivariance exclusive of each other or complementary? Additionally, when combined with the information-theoretic arguments regarding the benefits of symmetry breaking, what is the role of augmentations there?
- Is it reasonable to expect that these results and conclusions generalize to different types of groups? What would be some necessary conditions for these results to generalize?

**Ethical Concerns:**

["NO or VERY MINOR ethics concerns only"]

**Final Justification:**

This work provides an extensive investigation of the effect of equivariance which, although focused on point cloud processing tasks, offers valuable insights to the community. Additionally, during the rebuttal, the authors addressed all of the concerns I raised in my initial review. For these reasons, I maintain my rating of accept.

**Limitations:**

Yes

**Paper Formatting Concerns:**

No formatting concerns in the paper

**Quality:**

4

**Strengths And Weaknesses:**

Strengths:
- The topic of the paper is very timely, as understanding the benefits and the tradeoffs of utilizing equivariant networks is something that the geometric deep learning community is currently actively interested in. Additionally, the authors do a good job in the introduction of the paper of providing readers with the context surrounding the research questions this work aims to answer.
- The proposed architecture is very flexible and can be of interest to the community even beyond the investigation performed in this work.
- The experimental evaluation is extensive, showing how the different combinations of equivariant tasks and symmetry-breaking mechanisms can affect performance. This can be of great use to practitioners who want to decide the degrees of symmetries they want to apply to their specific task.

Weaknesses:
- While the evaluation of different degrees of equivariance is extensive, there is no discussion or comparison with the use of data augmentation, which is one of the most common ways of incorporating known symmetries into the learning process.
- The paper focuses on symmetries that are subgroups of SE(3). It is not clear if these conclusions generalize to symmetries of groups that are fundamentally different than SE(3).

---

> ### Author Rebuttal · Authors · 2025-07-31
>
> Dear reviewer, We thank you for the time and effort you have put into providing a review of our work. We sincerely thank you for your thoughtful and encouraging feedback. We are glad that you found the topic of the paper to be **timely and relevant** to ongoing discussions in the geometric deep learning community, particularly regarding the understanding of the benefits and trade-offs of equivariant networks. Your comments on the *flexibility* of the proposed architecture are greatly valued. We appreciate that you found our experimental evaluation **extensive**.
>
> In the text below, we try to address your concerns and respond to the questions:
>
> ##  Data Augmentation [W1]
>
> We would like to emphasise that all model variations for tasks that are supposedly equivariant **have rotation augmentations** for training- we will explicitly mention this in the experiments section.
>
> ## Equivariance beyond E3/SE3 [W2]
>
> The reviewer raises an important point about understanding the effects of equivariance beyond SE(3) and its subgroups. We would like to first justify our choices made in the paper by mentioning that we focus on standard point clouds in the real world that have mainly E(3) symmetries and permutation symmetries. Hence, we primarily end up focusing on E(3)/SE(3) and its subgroups.
>
> ## Response to Q1: Role of augmentation
>
> Generally, models without equivariant layers are trained with augmentations. So, we train with SO(3) augmentation for all models on tasks that are equivariant. For ShapeNet segmentation, we use standard augmentation (20° z-rotations + scaling), as full SO(3) augmentation hurts in-distribution performance. This helps us understand if models improve performance with augmentation, and they do. (*This effect is less in equivariant models than in non-equivariant models.*) We don’t see equivariance and augmentations as exclusive but complementary. With regards to symmetry breaking and augmentations, the non-equivariant method generally benefits from augmentations.
>
> ## Response to Q2: Generalization to other groups
>
> See W2. The reviewer raises an important point about the effects of equivariance and symmetry breaking for groups apart from SE(3) and its subgroups. We believe this is an important question that we aim to address in future work. As far as the usefulness of equivariance goes, as long as the task is equivariant, having equivariant layers is an effective way to learn symmetries for any group. Although the performance gap between non-equivariant and different equivariant groups depends on the **complexity** of the group [1].
>
> Once again, we thank the reviewer for the insightful feedback. If any concerns remain from your initial evaluation, we’d be happy to clarify further.
>
> References:
>
> [1] *On the hardness of learning under symmetries*, Kiani et al.

---

> > ### Comment · Reviewer_Bpqw · 2025-08-07
> > **Response to Rebuttal**
> >
> > I thank the authors for their rebuttal and for addressing most of my concerns.
> > I appreciate the clarification about the use of augmentations in the experimental evaluation of non-equivariant methods, and I thank the authors for adding it in the main text of this work.
> > Additionally I agree with the authors that while a more broad analysis of the effect of equivariance is important, since this work primarily focuses on point-cloud analysis tasks the focus on E(3)/SE(3) symmetry groups is sufficient.
> > Since the authors addressed my concerns I will retain my rate of accept.

---

> ### Comment · Area_Chair_q2ou · 2025-08-07
>
> Dear reviewer Bpqw, thanks for your time reviewing this paper. Although your opinion might be clear, according to this year's NeurIPS policy, it is required to have reviewer-author discussions before submitting the mandatory acknowledgement. Could you please take some time to comment on the authors' responses? Thanks!

---

### Decision · Program_Chairs · 2025-09-17

**Decision:**

Accept (poster)

**Comment:**

[Summary]

Based on the initial reviews as well as the reviewer-author discussions, my understanding of this paper's major contribution is two-fold: (i) introducing an architecture that unifies equivariant and non-equivariant models; (ii) (based on the proposed architecture,) investigating how different levels of equivariance affect model performances on various aspects.


[Strengths]

- The paper addresses an important research question in the field (timely, and large in scope).
- The paper does a good job studying this important topic, conducting extensive experiments and showing insightful analysis.
- In addition to the studies, the paper also proposes a practical framework unifying equivariant and non-equivariant models.
- The paper is well-written (in its background introduction, technical parts, and results presentation).


[Weaknesses]

It seems that the major concerns from the reviewers are not in the model design, but mostly in the experimental analysis/discussion part. Some major concerns are:
- at what scope can the statements/conclusions be drawn from the results;
- whether the key questions are properly asked/answered;
- whether the experimental designs and evaluation setups are solid enough (mostly on some details);
- and presentation of the experimental results.


[Reasons for decision]

The paper received two firm acceptances (Bpqw and mQdo), one borderline acceptance (KUZW, with remaining concerns in writing), and one borderline rejection (V9ah) in the end.

In my opinion, V9ah didn't show enough opinions or participation in the discussion to strongly suggest rejection. And given the following reasons:
- (i) the two initial acceptance reviews are firm;
- (ii) KUZW overweights the paper's contribution to the field's long-term discussions than its weaknesses;
- (iii) although there're remaining concerns on the writing (mostly on how to phrase the statements), the paper is clearly written in its technical parts (as the other reviewers commented this papaer as "well-written", and KUZW also mentioned that the background introduction was accessible to a broader audience); and the authors have committed to addressing KUZW's remaining concerns in writing in their final revision;
- (iv) for an experimental study paper on a long-existing open problem in the field, it may not be easy to answer all questions properly in one work (under a certain set of experimental designs), and I feel the remaining unclear questions are in an acceptable range;
I would like to recommend acceptance. But I hope the authors could further improve their writing based on the reviews in their final revision.


[Discussion summary]

Initially, the paper received diverging scores, with the acceptance opinions slightly stronger than the rejection opinions -- two reviewers (Bpqw and mQdo) gave accepts, and two reviewers (KUZW and V9ah) gave borderline rejects.

There weren't too many discussions. Some major discussions were with KUZW, where (i) most of KUZW's concerns were addressed, (ii) the remaining concerns were mostly on the writing (statements), and (iii) KUZW changed the score to acceptance in the end. And KUZW also mentioned that this paper can contribute to the long-running debate for inductive biases vs. scale for at least point clouds.